# Temporal analysis of melanogenesis identifies fatty acid metabolism as key skin pigment regulator

**Farina Sultan[1,2], Reelina Basu[1], Divya Murthy[1], Manisha Kochar[3], Kuldeep S. Attri[4], Ayush Aggarwal[1,2], Pooja Kumari[5], Pooja Dnyane[2,6], Jyoti Tanwar[1,2,7], Rajender K. Motiani[7], Archana Singh[1,2], Chetan Gadgil[1,2,6], Neel Sarovar Bhavesh[5], Pankaj K. Singh[4], Vivek T. Natarajan[1,2]\*, Rajesh S. Gokhale [3¤]\***

1 CSIR-Institute of Genomics and Integrative Biology, New Delhi, India, 2 Academy of Scientific and Innovative Research, Ghaziabad, UP, India, 3 Immunometabolism Laboratory, National Institute of Immunology, New Delhi, India, 4 Eppley Institute for Research in Cancer and Allied Diseases, University of Nebraska Medical Center, Omaha, Nebraska, United States of America, 5 International Centre for Genetic Engineering and Biotechnology, New Delhi, India, 6 CSIR-National Chemical Laboratory, Pune, Maharashtra, India, 7 Laboratory of Calciomics and Systemic Pathophysiology, Regional Centre of Biotechnology (RCB), Faridabad, Haryana, India

¤ Current address: Indian Institute of Science Education and Research, Pune, Maharashtra, India
* tnvivek@igib.in (VTN); rsg@nii.ac.in (RSG)

**Data Availability Statement:** The accession number for the RNA sequencing data reported in this paper is GSE164375 (https://www.ncbi.nlm.nih.gov/geo/query/acc.cgi?acc=GSE164375).

## Abstract

Therapeutic methods to modulate skin pigmentation has important implications for skin cancer prevention and for treating cutaneous hyperpigmentary conditions. Towards defining new potential targets, we followed temporal dynamics of melanogenesis using a cell-autonomous pigmentation model. Our study elucidates 3 dominant phases of synchronized metabolic and transcriptional reprogramming. The melanogenic trigger is associated with high MITF levels along with rapid uptake of glucose. The transition to pigmented state is accompanied by increased glucose channelisation to anabolic pathways that support melanosome biogenesis. SREBF1-mediated up-regulation of fatty acid synthesis results in a transient accumulation of lipid droplets and enhancement of fatty acids oxidation through mitochondrial respiration. While this heightened bioenergetic activity is important to sustain melanogenesis, it impairs mitochondria lately, shifting the metabolism towards glycolysis. This recovery phase is accompanied by activation of the NRF2 detoxication pathway. Finally, we show that inhibitors of lipid metabolism can resolve hyperpigmentary conditions in a guinea pig UV-tanning model. Our study reveals rewiring of the metabolic circuit during melanogenesis, and fatty acid metabolism as a potential therapeutic target in a variety of cutaneous diseases manifesting hyperpigmentary phenotype.

## Introduction

Despite significant progress in understanding the physiology and biochemistry of human skin pigmentation, the strategies to manipulate this phenomenon for clinical benefit has met with

Further, the authors declare that all other data supporting the findings of this study are available within the paper and Source Data files.

**Funding:** R.S.G. acknowledges support from J.C. Bose fellowship and Department of Biotechnology (DBT) for providing funds to National Institute of Immunology (SB/S2/ JCB-038/2015). R.S.G., T.N. V. & C.G. acknowledge the support of Council for Scientific and Industrial Research (CSIR) project TOUCH (BSC0302). R.K.M. acknowledges support of Department of Biotechnology/Wellcome Trust India Alliance Intermediate Fellowship (IA/I/19/2/ 504651). F.S. acknowledges UGC JRF/SRF fellowship. D.M. and K.S.A. were CSIR JRF/SRF Fellows. M.K. was supported by funds from SERB project. P.K., P.D. & A.A. are recipients of the CSIR JRF/SRF fellowship. The funders had no role in study design, data collection and analysis, decision to publish, or preparation of the manuscript.

**Competing interests:** I have read the journal's policy and the authors of this manuscript have the following competing interests: R.S.G. is a Co-founder of Vyome Biosciences Pvt. Ltd., a biopharmaceutical company working in the Dermatology area. Part of the study is patented under the Indian Patent Act. CSIR-IGIB and NII jointly applied for the patent titled "Compositions having application against hyper-pigmentation". Inventors are listed as Farina Sultan, Manisha Kochar, Rashmi Sanjay Bhosale, Vivek T. Natarajan and Rajesh S. Gokhale. Indian Patent Application No. 202011047316. Filing Date: 29.10.2020. Other authors do not have any conflict of interest.

**Abbreviations:** BCA, bicinchoninic acid; BSA, bovine serum albumin; BMRB, Biological Magnetic Resonance Bank; CIP, calf intestinal phosphatase; DAG, diacylglycerol; DEG, differentially expressed gene; ER, endoplasmic reticulum; ETC, electron transport chain; HBP, hexosamine biosynthesis pathway; HMDB, Human Metabolome Database; HSL, hormone-sensitive lipase; LRT, likelihood-ratio test; MC1R, melanocortin 1 receptor; MiNA, Mitochondrial Network Analysis; NMR, nuclear magnetic resonance; O-GlcNAc, O-linked β-N-acetylglucosamine; PCA, principal component analysis; PDH, pyruvate dehydrogenase; PDK1, pyruvate dehydrogenase kinase; PPP, pentose phosphate pathway; PTU, 1-phenyl-2-thiourea; qRT-PCR, quantitative real-time polymerase chain reaction; ROS, reactive oxygen species; siRNA, small interfering RNA; TAG, triacylglycerol; TCA, tricarboxylic acid cycle; TEM, transmission electron microscopy; TF, transcription factor; TG, target gene; UDP-GlcNAc, uridine diphosphate-β-N-acetylglucosamine; VST, variance-stabilizing transformation; 25-HC, 25-hydroxycholesterol.

minimal success. To counter the deleterious effects of UV radiations, human skin activates melanisation that protects skin from cancer and photoaging [1]. On the other hand, pathological hyperpigmentation response of skin occurs due to inflammatory conditions [2]. Patchy cutaneous hyperpigmentation is an associated comorbidity in more than 30% of patients with diabetes and obesity [3]. Skin pigmentation is due to the presence of melanocyte cells in the epidermis, which possesses biosynthetic machinery to produce melanin within melanosomes [4]. These specialized membrane-bound organelles are then transferred to neighbouring keratinocytes imparting photoprotection [5,6]. Melanogenesis thus can be considered to be a conglomeration of many interacting components wherein individual components, as well as the interaction networks, manifest spatiotemporal coherence. Perturbations within any of these events result in homeostatic imbalance leading to a disease phenotype. The challenge is to elucidate dynamic temporal interactions between constituent molecules of various cellular processes.

MITF, the central transcription regulator of melanocyte lineage, connect array of gene networks pertaining to melanogenesis, proliferation, and survival [7]. MITF regulates the expression of numerous pigmentation-associated genes such as PMEL17 and MART1 and melanin synthesis enzymes TYR, DCT, and TYRP1 to promote melanocyte differentiation [8–10]. UV-mediated activation of pigmentation proceeds through secretion of α-MSH by keratinocytes, which binds to epidermal melanocytes receptor, the melanocortin 1 receptor (MC1R), triggering cAMP production and CREB-mediated MITF transcription [11–13]. Further, coactivators like SOX10 activate MITF, while suppressors like TCF4 and ATF4 down-regulate MITF transcriptional response [14–16]. Melanocytes thus possess the ability to respond to environmental signals and assume a wide variety of distinct functional fates. These specialized cells of the epidermis return to a resting state, where they are known to persist, potentially prepared for another round of activation. These distinct phases of melanocytes can be anticipated to be dependent on dynamic changes in cellular metabolism to cater cellular energy demands and biomolecule requirements.

Several lines of evidence suggest the alteration of mitochondrial function during melanogenesis, some of which are rather paradoxical. For example, induction of melanin synthesis in B16F10 melanoma cells reduces the oxygen consumption after 48 hours of stimulation, without changes in mitochondrial membrane potential [17]. However, mitochondrial mass is reported to be higher in cells with melanogenesis stimulation [17]. A fraction of these mitochondria is shown to be in direct contact with melanosomes, where the interorganelle connections mediated by fibrillar bridges are proposed to facilitate the exchange of small molecules between the 2 organelles [18]. Remodelling mitochondria towards increased fission enhances reactive oxygen species (ROS) levels that can have contradictory effect on melanogenesis [19,20]. While targeting $F_1F_0$-ATP synthase that should also result in ROS accumulation, induces hyperpigmentation [21]. In a recent study, untargeted metabolomics of α-MSH-induced B16F10 cells analysed at 1, 24, and 48 hours showed minimal changes in the metabolite profile, when compared with their respective controls [22]. While it is surprising, a possibility is that the time points used in this study do not capture the dynamics of metabolite concentrations and metabolic fluxes. Another distinct possibility is the heterogeneity within cellular populations and cell surface receptors, which could obscure the interpretation of the results. Cellular heterogeneity has been reported for primary melanocyte cultures, where the cells could be transiting between precursor cells and their descendants at different stages of pigmentation [23]. For the identification of transient regulatory events during melanogenesis, it is pertinent to resolve melanocyte function over time in a synchronized model system that can capture a full array of events.

As metabolic reprogramming is emerging as a hallmark of cellular effector functions, it is important to leverage metabolic dependencies as a possible target for modulating skin pigmentation. In this study, we have employed a previously developed B16 cell-autonomous pigmentation model where cells transit from basal depigmented to the pigmented state over a period of 6 days [24]. Transcriptomic and metabolomic studies allowed us to identify dynamically changing key transcriptional network modules and corresponding metabolic configuration in a time-dependent manner. Along with defining a framework for understanding melanogenesis programming, our studies identify the transient, yet the key role of SREBF1-mediated fatty acid metabolism during the melanogenic phase. Based on the guinea pig tanning model, we show that inhibitors of fatty acid metabolism can resolve hyperpigmentary conditions, thus revealing new targets for modulating skin pigmentation.

## Results

### Melanogenesis is coupled to transcriptional activation of metabolic genes

To understand the differentiation programming of melanocytes from depigmented to pigmented state, we perform a global transcriptomic analysis of B16 cell-autonomous pigmentation model [24]. B16 pigmentation model is a density-dependent model, wherein cells are seeded at a very low density of 100 cells/cm$^2$ and transition from basal depigmented to the pigmented state of cells occurs over a period of 6 days as shown in the schematic (Fig 1A). In this model, melanogenesis is probably triggered by the fine balance between the intrinsic needs of the cells and the constraints imposed by the extrinsic conditions. A series of coordinated processes encompassing transcriptional activation, melanosome biogenesis and melanin synthesis can be captured at the phenotypic and molecular level. The phenotypic changes in the melanin are best observed and quantitated from days 3 to 6 (Fig 1A) [25]. At the molecular level, we observe a higher MITF protein expression on days 3 and 4. The classical MITF-mediated pigmentation targets show different trajectories of expressions during this period. PMEL17, a marker of early-stage melanosomes, is found to peak around days 4 and 5, indicating new melanosomes formation as an early event that diminishes by day 6. This is followed by increased expression of tyrosinase, the rate-limiting enzyme in melanin synthesis, on days 5 and 6 (Fig 1B and 1C).

To study temporal events during melanogenesis, genome-wide transcriptome analysis was performed from days 3 to 6. Principal component analysis (PCA) of the transcriptome data showed separation of different samples on the major PC axis (PC1) (S1A Fig). The correlation coefficient "r" between the biological replicates of the same sample were in the range of 0.98 to 0.99 (S1B Fig), suggesting overall high concordance between the 2 replicates. Time-course analysis was performed using the LRT (likelihood-ratio test), and genes with adjusted *p*-value < 0.001 were taken as significant differentially expressed genes (DEGs) from days 3 to 6 using DESeq2. Heatmap was plotted for these 1,493 DEGs obtained in time-course analysis (Fig 1D). As expected, several key pigmentation-related genes like *Mitf*, *Pmel*, and *Tyr* could be observed among the DEGs. Hierarchical clustering was performed on the expression data of 1,493 DEGs, and the data were divided into 7 clusters. In general, genes on days 3 and 4 show substantially similar gene expression values as compared to days 5 and 6. Pathway enrichment analysis using the KEGG database, illustrated as Bubble Plot (Fig 1E), for these gene clusters revealed pathways like RNA transport, ribosome biogenesis, and spliceosome enriched on days 3 and 4, indicative of transcriptional activation during early pigmentation phases. Day 5 showed up-regulation of metabolic pathways like steroid biosynthesis, unsaturated fatty acid synthesis, and fatty acid metabolism. Enrichment of fatty acid metabolism

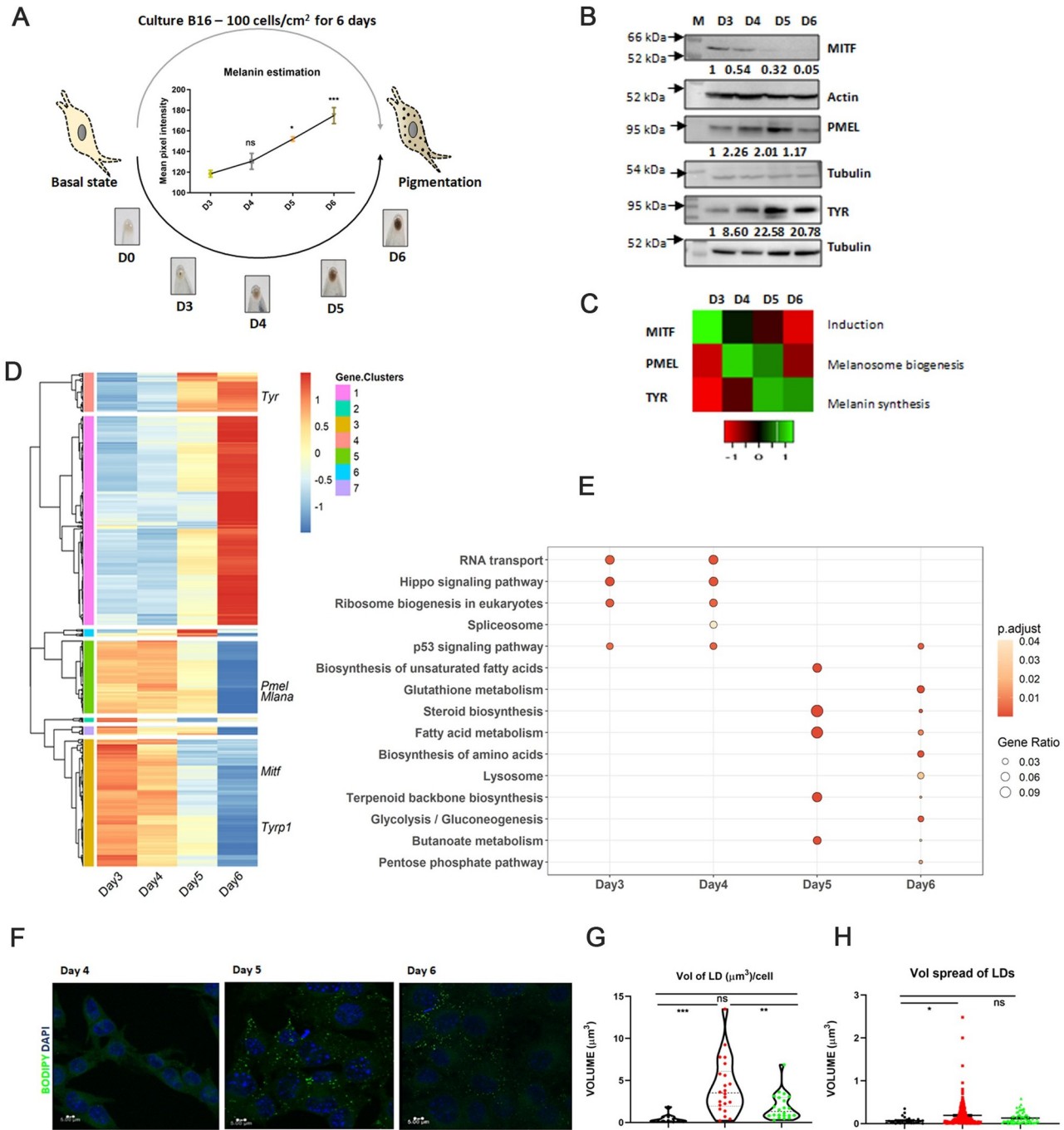

**Fig 1. Melanogenesis is coupled to transcriptional activation of metabolic genes.** (A) Schematic diagram illustrating the assay setup where culturing B16 cells at low density of 100 cells/cm$^2$ (Top) results in the transition of melanocyte from depigmented (day 0) to pigmented state (day 6) as shown in the pellet images ($N = 3$) (Bottom). Quantitative analysis of melanin intensity of the pellets is shown in the line graph (Between). Mean ± SEM is plotted for 3 biological replicates. One-way ANOVA is applied, F(3,8) = 18.28. $p$-Value = 0.0006. Turkey's test is performed for pairwise comparison. (B) Representative western blot of key pigmentation proteins, MITF, PMEL, TYR, and DCT on days 3 to 6 (D3 to D6) ($N = 3$). Arrows mark the sizes of molecular weight markers written in kDa. Numerical values show average fold change for 3 biological replicates. (C) Heatmap depicting relative mean expression levels of pigmentation proteins wrt D3 for 3 independent experiments. Scale from red to green represents the z-scores of fold change (below). (D) Heatmap represents 1,493 DEGs with an adjusted $p$-value < 0.001 obtained from time-course analysis performed using the LRT test for 2 independent biological replicates. Scale from blue to red represents z-score from −1 to +1. Hierarchical clustering was performed on the expression data of 1,493 DEGs into 7 clusters. (E) KEGG pathway enrichment analysis was done for DEGs with adjusted $p$-value < 0.001 up-regulated on days 3, 4, 5, and 6. Bubble plot depicts the enrichment of pathways on different days, where the size of bubble represents the gene ratio and colour represents the $p$-value. (F) Representative confocal microscopy images show lipid droplet accumulation in B16 cells during pigmentation. Images were taken at

63×. Scale is 5 μm. (**G**) Violin plot depicting the quantitation of total volume of lipid droplets per cell using VOLOCITY software. Approximately 25 to 30 cells are taken in each replicate. Mean ± SEM is plotted in 3 independent biological replicates. One-way ANOVA is applied, F(2,52) = 11.45. *p*-Value < 0.0001. Turkey's test is performed for pairwise comparison. ***p*-Value = 0.0001, **p*-value = 0.034. ns is not significant. (**H**) Dot plot depicting the quantitation of size of individual lipid droplets (as volume spread of lipid droplets) using VOLOCITY software. Approximately 25 to 30 cells are taken in each replicate. Mean ± SEM is plotted in 3 independent biological replicates. One-way ANOVA is applied, F(2,262) = 3.54. *p*-Value = 0.0303. Turkey's test is performed for pairwise comparison. Quantitative data are provided in S1 Data for Panels A, C, G, and H. Quantitative data for Panels D and E are provided in S3 Data. DEG, differentially expressed gene; LRT, likelihood-ratio test.

along with the up-regulation of glycolysis and glutathione metabolism could be noted on day 6.

An important facet of lipid metabolism is the assimilation of free fatty acids in the form of neutral lipids (triacylglycerols (TAGs)) within lipid droplets, which are otherwise toxic to cells. We analysed the expression of fatty acid synthesis genes and TAG synthesis genes from RNA sequencing data (S1C and S1D Fig). Analysis showed a concordant increased expression of several of the genes in both fatty acid and TAG synthesis pathways with pigmentation. This may result in increased lipid droplets formation during pigmentation. We, therefore, traced lipid droplets during pigmentation using BODIPY dye (Fig 1F). Quantitative analysis of the volume of lipid droplets per cell and volume of individual lipid droplets revealed the formation of these lipid aggregates significantly increases on D5 (Fig 1G and 1H). Further, a significant and rapid depletion in the volume of lipid droplets per cell was observed on D6. These results suggest that B16 cells assimilate and utilise fatty acids synthesized by cells as lipid droplets.

To demonstrate that these metabolic changes are a consequence of pigmentation, we treated the B16 cells grown at low density with 1-phenyl-2-thiourea (PTU), which inhibits the formation of melanin pigment. PTU-treated cells show depigmented phenotype and, in fact, exhibit enhanced proliferation, when compared with pigmented day 6 cells (S1E and S1F Fig). Examination of FASN in PTU-treated cells showed a significant decrease in protein expression, contrary to the increase observed for pigmenting B16 cells (S1G–S1J Fig). These studies provide credence to the hypothesis that the synchronized modulation of metabolic pathways identified during transcriptome studies is associated with cellular pigmentation.

## Differential transcriptionally regulated metabolic networks emerge at different stages of melanogenesis

To identify prominent transcriptional networks on days 3 to 6, we performed transcription factor (TF) enrichment analysis for differentially regulated genes using the TRRUST database. We constructed TF-target gene (TG) network maps for the top 7 TFs identified on each day. This was overlaid with transcript expression changes observed for the TGs (Fig 2). Such a temporal representation of the TF-TG network maps revealed an interesting insight into the transcriptional programming during the course of pigmentation. On day 3, pigmentation regulators *Mitf* and *Egr1* networks are functional, both of which are known to induce pigmentation response in melanocytes [26,27]. Another melanogenesis-associated gene *Tcf4* could also be noted in the days 3 and 4 regulons, which is known to suppress *Mitf* levels [15]. The expression of *Myc* network, which is involved in proliferation [28], decreases with each day and this is congruent with our experimental observations. On day 5, completely new sets of TF-TG networks emerge, prominent of them are the *Srebf1* and *Srebf2* clusters, which are known to regulate lipid metabolism across different cell types [29]. Another significant cluster that becomes evident is the *Nfe2l2 (Nrf2)* regulon. This TF is involved in the phase II detoxification pathway, and previous studies have shown that melanocytes resist oxidative detoxification through a robust expression of this pathway during pigmentation [30]. While the transcriptional analysis provides interesting insights into the temporal regulation of TF-TG

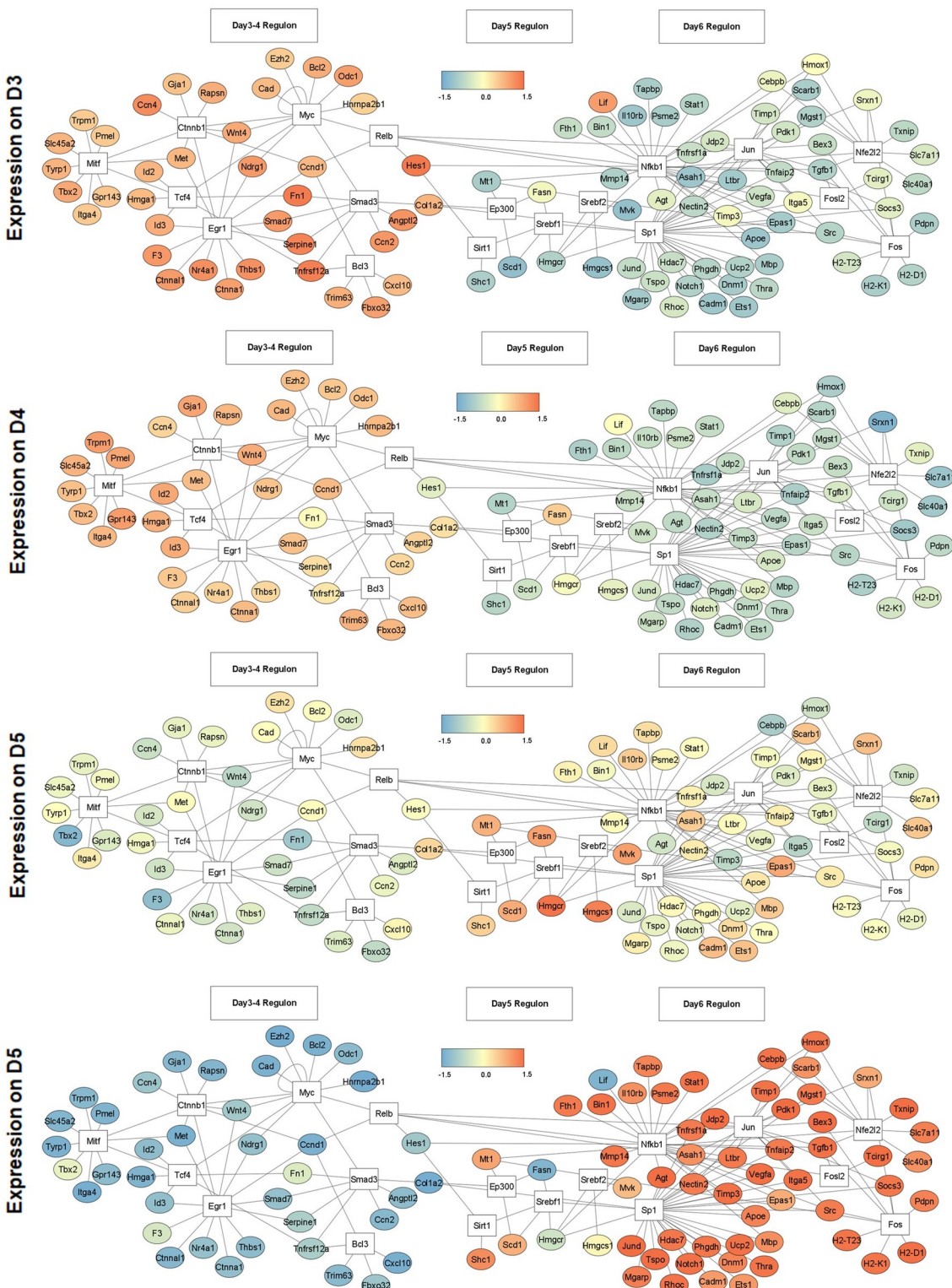

**Fig 2. Transcriptionally regulated metabolic networks map for different stages of melanogenesis.** DEGs were analysed for the TG enrichment of TF using the TRRUST database on metascape (metascape.org). TF-TG network analysis shows top 7 up-regulated TFs obtained in DEGs analysis on days 3 to 6, along with the expression profile of their TGs. TFs are enclosed in white box and the lines connect each TF to the set of genes regulated by them during pigmentation. TGs are enclosed in circle and colour gradient represents the scaled (row-wise) expression values from the RNA Seq data. Quantitative data are provided in S3 Data. DEG, differentially expressed gene; TF, transcription factor; TG, target gene.

networks, we mainly focus on the metabolic programming of these cells during different phases of melanogenesis.

To validate key transcriptional networks that emerged on different days in TF-TG analysis, we analysed mRNA expression of *Mitf*, *Srebf1*, *Nrf2*, and their downstream TGs. Consistent with the TF-TG analysis, *Mitf* showed significant up-regulation on days 3 and 4, (S2A Fig), and *Tyr*, a rate-limiting enzyme in melanin synthesis, showed significantly higher expression on days 5 and 6 (S2B Fig). *Srebf1* emerges as a prominent metabolic network on day 5. *Srebf1* showed maximum up-regulation on day 5, followed by day 6 at mRNA level (S2C Fig). This is congruent with the observed induction up to 1.5- to 2-fold for key fatty acid synthesis genes, *Fasn*, *Acaca*, *Acacb*, and *Acly* on days 5 and 6, as analysed by quantitative real-time polymerase chain reaction (qRT-PCR) (S2D Fig). Another important network that emerged on day 6 is *Nrf2*. Examination of mRNA expression of *Nrf2* and its TGs, *Gsr* and *Gst*, showed a consistent up-regulation on day 6 (S2E and S2F Fig). Together, TF-TG analysis delineates diverse transcriptional networks regulating metabolism, mediating an important role during the transition from depigmented to pigmented state.

## Steady-state analysis of polar metabolites using mass spectrometry during pigmentation

To understand how metabolic changes are linked with the process of melanogenesis, we measured the levels of polar metabolites from days 3 to 6 cells using liquid chromatography-coupled tandem mass spectrometry. A total of 306 peaks were obtained, out of which 175 were mapped to different metabolites with high confidence. The relatedness between all the data sets was compared using PCA. All replicates showed good clustering and clear day-wise segregation was observed (S3A Fig). Hierarchical clustering analysis based on the top 50 regulated metabolites showed an increased level of metabolites corresponding to nucleotide and amino acid metabolism during days 3 and 4. On days 5 and 6, a substantial number of cofactors and Kreb's cycle metabolites were seen to be up-regulated (S3B Fig). To understand the pathway-based connectivity, we overlaid the data on 4 major central carbon metabolic pathways—glycolysis, tricarboxylic acid cycle (TCA), pentose phosphate pathway (PPP), and hexosamine biosynthesis pathway (HBP) (Fig 3A). Analysis of glycolytic intermediates showed 2 distinct patterns of regulation. Metabolites in the upper half of the glycolysis (6-carbon metabolites) showed limited variations across all 4 days. The upper half of glycolysis is known to branch into the PPP and HBP. Indeed, analysis of metabolites from PPP and HBP indicated an increased accumulation on days 5 and 6. PPP generates NADPH to maintain a redox environment and provides intermediates for fatty acid and nucleotide synthesis. HBP forms uridine diphosphate-β-N-acetylglucosamine (UDP-GlcNAc) moiety required for O-linked β-N-acetylglucosamine (O-GlcNAc) posttranslational modification of melanogenic proteins [31,32]. The metabolites in the lower half of glycolysis (3-carbon metabolites) increased approximately 4-fold during the pigmentation phase. The lower half of the glycolysis produces lactate and feeds into the TCA cycle. Interestingly, the levels of both lactate and TCA metabolites show an increase with time. Melanogenesis trigger augments distribution of glucose into all the branches of central carbon metabolism during pigmentation, providing necessary metabolic pools, probably required for melanosome formation and maturation within melanocytes. Furthermore, increased HBP and PPP metabolic pathways suggest a supporting role for the synthesis of a variety of biomolecules.

## [U-¹³C]-Palmitate incorporation in TCA metabolites during pigmentation phase

To understand how the glucose uptake during the pigmentation phase (days 5 and 6) is quantitatively apportioned between the lactate and TCA metabolites, we performed stable isotope

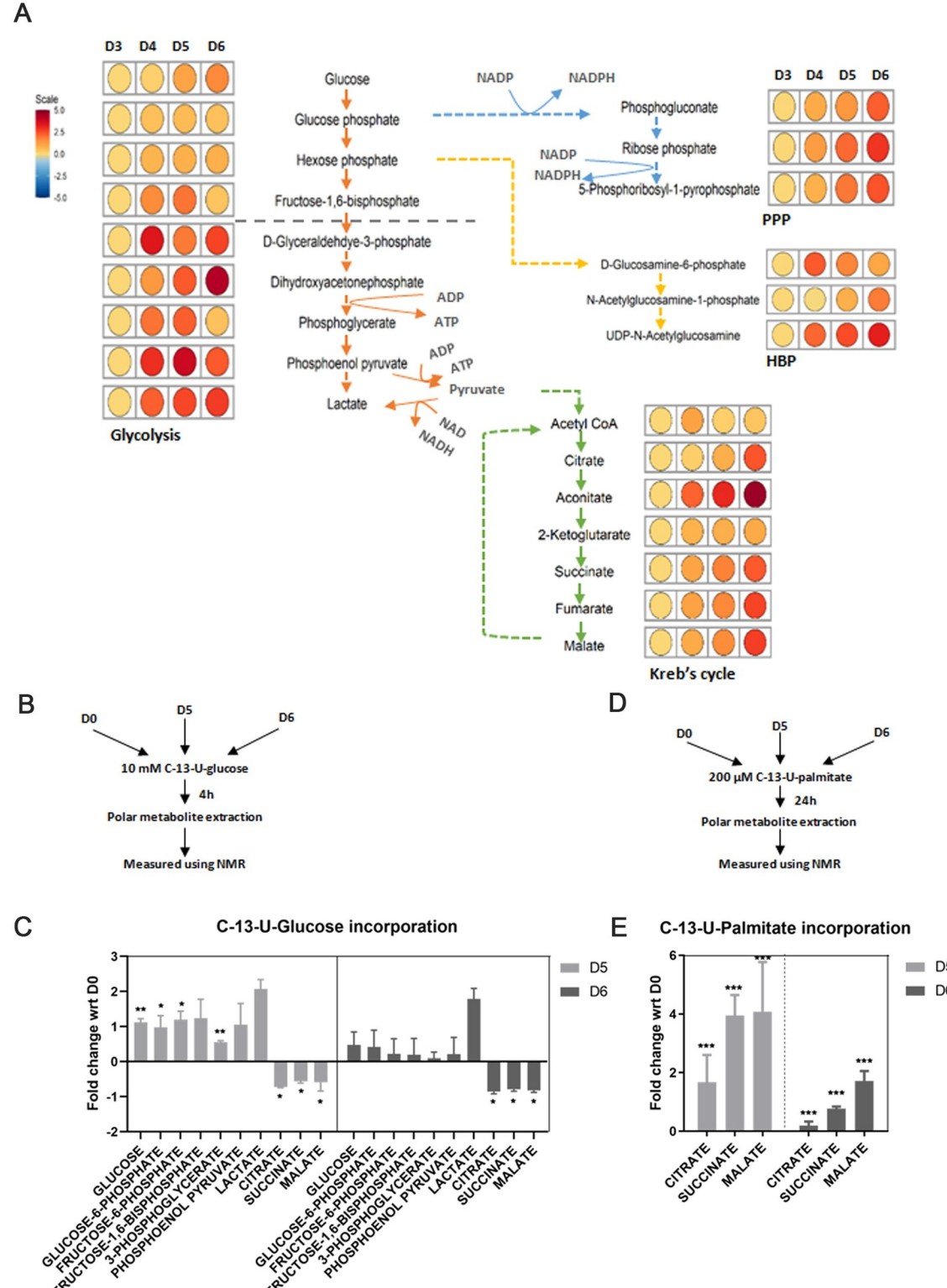

**Fig 3. Partial uncoupling of glycolysis and TCA supports biosynthesis pathways (PPP, HBP) while FAO increases to sustain TCA. (A)** Liquid chromatography-coupled tandem mass spectrometry-based metabolomic analysis is performed for polar metabolites extracted from D3 to D6 for 4 independent replicates. Metabolic network is drawn for glycolysis, Kreb's cycle, PPP, and HBP by comparing their metabolite levels across D3 to D6. Fold changes are plotted as heatmaps along each metabolite in the pathway with respect to D3. Scale bar ranges from blue to red corresponding to metabolite change from −5 to +5 fold. **(B)** Schematic

depicting study design for pulse-chase labelling of uniformly $^{13}$C labeled glucose in glycolysis and TCA metabolites at D0, D5, and D6 using NMR. (**C**) NMR-based quantitation of [U-$^{13}$C]-glucose incorporation in glycolysis and TCA metabolites at D5 and D6 with respect to D0. Mean ± SEM is plotted for 3 independent replicates. Two-way ANOVA is applied, time factor, $F(1.667,33.33) = 33.95$, $p$-value < 0.0001. Dunnett's test is performed for pairwise analysis. $^{***}p$-Value < 0.005, $^{**}p$-value < 0.01, $^{*}p$-value < 0.05. ns is not significant. (**D**) Schematic depicting study design for pulse-chase labelling of uniformly $^{13}$C labeled palmitate in TCA metabolites at D5 and D6 with respect to D0 using NMR. (**E**) NMR-based quantitation of [U-$^{13}$C]-palmitate incorporation in TCA metabolites at D5 and D6. Mean ± SEM is plotted for 3 independent replicates. Two-way ANOVA is applied, time factor, $F(2,12) = 19.91$, $p$-value < 0.0002. Dunnett's test is performed for pairwise analysis. $^{***}p$-Value < 0.005, $^{**}p$-value < 0.01, $^{*}p$-value < 0.05. ns is not significant. Quantitative data are provided in S1 Data for Panels C and E. Quantitative data for Panel A are provided in S4 Data. FAO, fatty acid oxidation; HBP, hexosamine biosynthesis pathway; NMR, nuclear magnetic resonance; PPP, pentose phosphate pathway; TCA, tricarboxylic acid cycle.

tracing using labelled [U-$^{13}$C]-Glucose. These pulse-labelling experiments were analyzed after 4 hours of addition of 10 mM [U-$^{13}$C]-Glucose in the glucose-free medium (Fig 3B). The incorporation of $^{13}$C isotope in different metabolites was followed by nuclear magnetic resonance (NMR)-based measurements and several metabolites could be identified with high confidence. We observed a 2-fold increase in the $^{13}$C labelling of glycolytic metabolites on day 5 when compared with depigmented cells (Fig 3C). In contrast, the TCA metabolites—citrate, succinate, and malate showed a substantial decrease in the $^{13}$C incorporation. This configuration of increased glycolytic metabolites and decrease in TCA metabolites is also observed for day 6, suggesting that the pigmentation phase is associated with decreased channelisation of pyruvate to acetyl CoA. Interestingly, incorporation of $^{13}$C labelling on day 6 in glycolytic metabolites is lower than day 5, while relatively little variation is observed for TCA metabolites and lactate levels. This suggested an increased uncoupling of glycolysis and TCA. The enhancement in glycolytic flux on days 5 and 6 is majorly contributed by increased utilization of glucose from media (S3C Fig).

Earlier studies have shown the role of pyruvate dehydrogenase kinase (PDK1) in reducing the activity of pyruvate dehydrogenase (PDH), which is responsible for converting pyruvate to acetyl-CoA [33,34]. We observed increased expression of *Pdk1* during the late phase of pigmentation, which substantiates the quantitative increase in the level of lactate formed at day 6 (S3D–S3F Fig). Overall, this indicates the uncoupling of glycolysis and TCA in the late recovery phase of pigmentation. Incidentally, steady-state analysis had suggested an increase in the accumulation of TCA metabolites during day 6. We, therefore, reasoned that other pathways may be directly feeding into TCA metabolites.

Since up-regulation of fatty acid synthesis and metabolism was observed in transcriptional analysis, we evaluated the role of fatty acid oxidation in generating acetyl-CoA pools by following incorporation of [U-$^{13}$C]-Palmitate into TCA metabolites during the pigmentation phase (Fig 3D). NMR measurements of $^{13}$C label revealed a more than 2-fold increase in the incorporation on day 5 when compared with depigmented cells (Fig 3E). A subsequent decrease was observed on day 6, suggesting a restricted requirement of oxidative phosphorylation in maintaining the energy demands of the cells in the late phase of pigmentation. Such a dynamical metabolic adaptation during the pigmentation phase may be a crucial metabolic switch that shunts fatty acids to mitochondria to maintain ATP production and sustain PPP via glycolytic shunting to keep a balance between available nutrients and energy.

## Fatty acid availability increases mitochondrial respiration during pigmentation

To determine how cellular bioenergetics is affected by the availability of carbon source, we measured oxygen consumption rate in either glucose- or oleate-supplemented media using the Seahorse Mito Stress Test assay (Fig 4A). We observed that the basal respiration rate, which

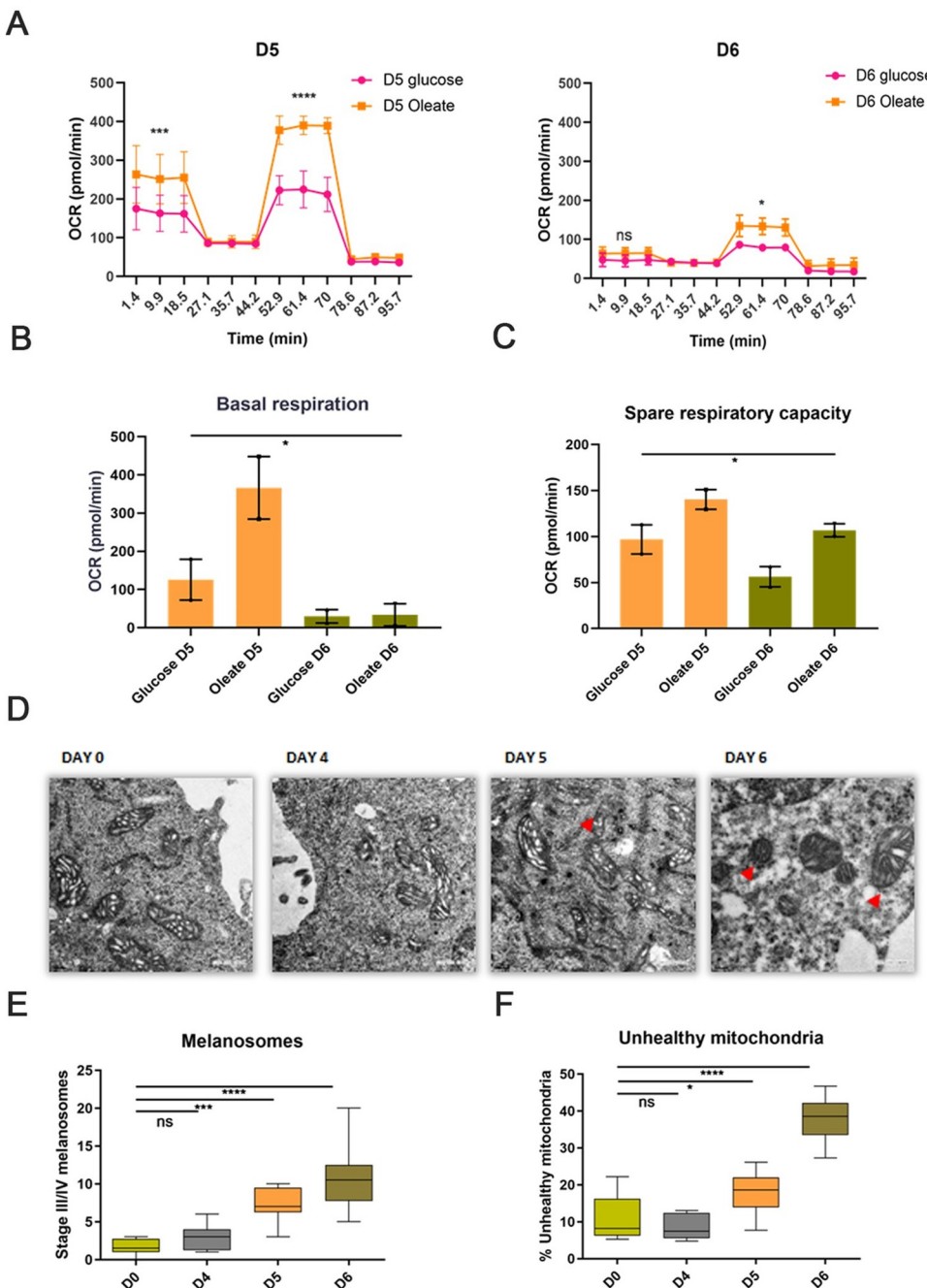

**Fig 4. Fatty acids are preferred carbon source for mitochondrial respiration during pigmentation. (A)**
Comparative analysis of oxygen consumption rate in glucose or fatty acid supplemented media on D5 and D6 using
Seahorse Mito Stress Test assay. Represented plot depicts Mean ± SEM for 2 independent biological replicates with 3
technical replicates in each set. Two-way ANOVA is applied. Sidak's multiple comparison test is performed for
pairwise analysis. *p*-Value. ****$p$-value < 0.001, ***$p$-value < 0.005, **$p$-value < 0.01, *$p$-value < 0.05. ns is not
significant. (**B**) Quantitative analysis of basal respiration in glucose or fatty acid supplemented media on D5 and D6.
Mean ± SEM is plotted for 2 biological replicates with 3 technical replicates for each set. One-way ANOVA is applied F
(3,4) = 9.337, *p*-value = 0.028. (**C**) Quantitative analysis of spare respiratory capacity in glucose or fatty acid
supplemented media on D5 and D6. Mean ± SEM is plotted for 2 biological replicates with 3 technical replicates for
each set. One-way ANOVA is applied F (3,4) = 8.915, *p*-value = 0.0303. (**D**) Representative TEM-based analysis
showing mitochondrial cristae morphology change during pigmentation from D3 to D6 (Number of cells, *n* = 8).
Images were taken at 3500× magnification. Scale is 0.2 μm (Bottom left). (**E**) Box plot depicting quantitative analysis of
stage III/IV melanosomes on different days (Number of cells, *n* = 8). Whiskers represent min and max range with

mean as centre line. One-way ANOVA is applied F(3,28) = 19.60, $p$-value < 0.0001. Tukey's test is performed for pairwise comparison. ****$p$-Value < 0.0001, ***$p$-value = 0.007. ns is not significant. (**F**) Box plot depicting quantitative analysis of unhealthy mitochondria on different days (Number of cells, $n$ = 8). Whiskers represent min and max range with mean as centre line. One-way ANOVA is applied F(3,28) = 47.08, $p$-value < 0.0001. Tukey's test is performed for pairwise comparison. ****$p$-Value < 0.0001, *$p$-value = 0.027. ns is not significant. Quantitative data are provided in S1 Data for Panels A, B, C, E, and F. TEM, transmission electron microscopy.

accounts for total oxygen consumption by the cells, is higher on day 5 than on day 6 (Fig 4B). However, in the presence of oleate, cells show higher basal respiration on day 5 in comparison to glucose. Availability of oleate in the media also results in higher spare respiration capacity of

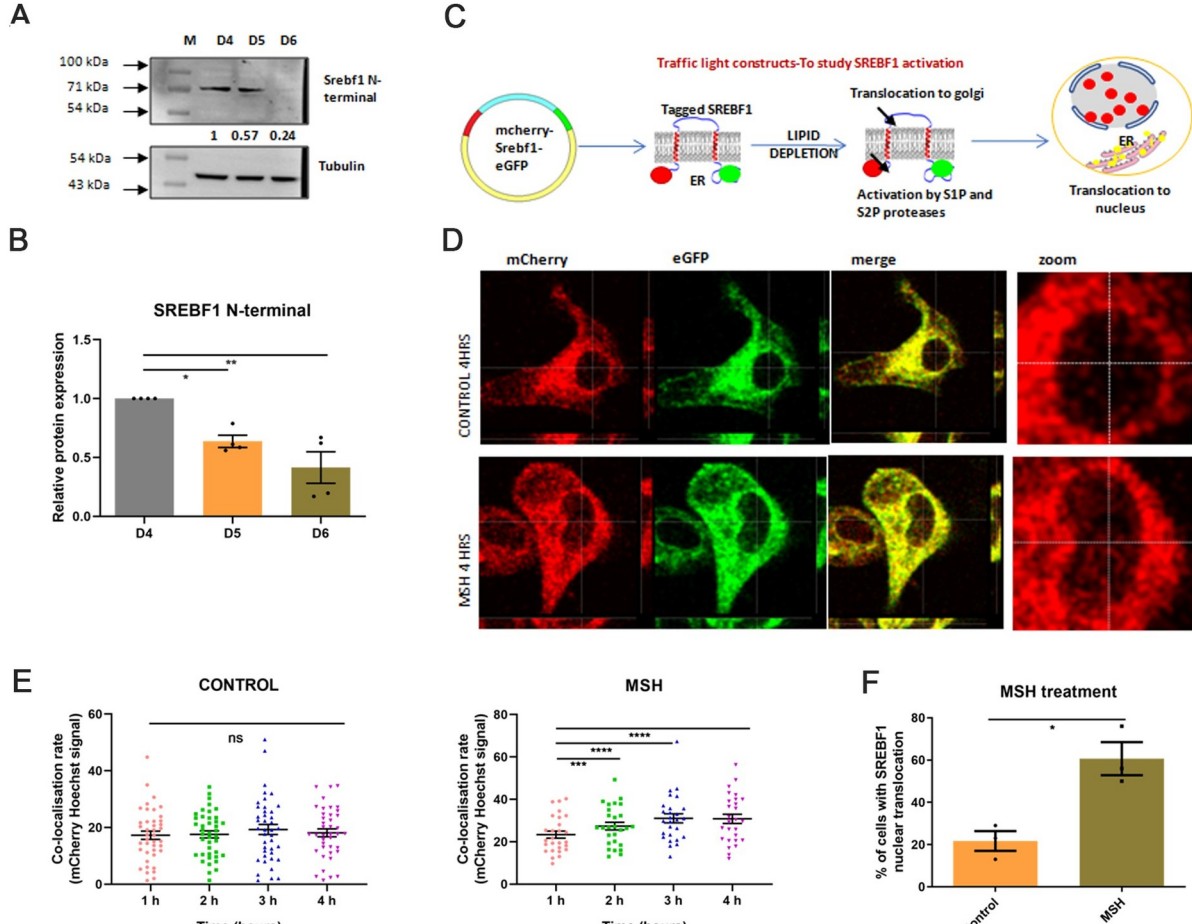

Fig 5. Melanogenesis induction causes activation of SREBF1. (**A**) Representative western blot showing SREBF1 N-terminal levels on different days of pigmentation, normalized to tubulin ($N$ = 4). Numerical values show average fold change for 4 biological replicates. (**B**) Bar graph depicting quantitative fold change of Srebf1 N-terminal expression with respect to D4. Mean ± SEM is plotted for 4 biological replicates. One-way ANOVA is applied, F (2,9) = 12.71, $p$-value = 0.0024. (**C**) Schematic design for studying SREBF1 activation using traffic light mCherry-Srebf1-eGFP vector and analyzing localization in different cellular compartments. Full-length protein localized in ER and gives yellow signal due to colocalization of mCherry and eGFP fluorescence. Cleaved SREBF1 N-terminal is transported to the nucleus along with mCherry tag, thus, accumulation of mCherry signal in the nucleus is indicative of SREBF1 activation. (**D**) Representative fluorescence images showing mCherry, eGFP, and merge signal. Magnified nuclear images were shown to focus on SREBF1 nuclear translocation after 4 hours of MSH treatment. DMSO is taken as vehicle control. (**E**) Dot plot depicts colocalization rate between mCherry and Hoechst signal analyzed for each cell at 1 to 4 hours, for control and MSH treatment. Around 20 to 30 cells were analyzed in each of the 3 biological replicates. One-way ANOVA is applied. For DMSO, F(2.512,99.63) = 0.9264, $p$-value = 0.4175, for MSH F(2,668,39.36) = 15.14, $p$-value < 0.0001. Dunnett's multiple comparison test is performed for pairwise analysis. ****$p$-Value < 0.0001, ***$p$-value < 0.0009. ns is nonsignificant. (**F**) Bar graph depicting the quantitation of number of cells showing positive phenotype after MSH treatment, determined by increased colocalization rate of mCherry and Hoechst signal from 1 to 4 hours. Mean ± SEM is plotted for combined analysis. Two-tailed Student $t$ test is performed, t = 4.267, df = 4 $p$-value = 0.0130. Quantitative data are provided in S1 Data for Panels B, E, and F.

cells on both days, days 5 and 6 (Fig 4C). This indicates that fatty acids rapidly undergo β-oxidation resulting in increased electron flow through mitochondrial complexes on day 5. Together, these studies provide support to the hypothesis that glucose is partially uncoupled from TCA while fatty acid oxidation is the preferred pathway during the intermediate phase of pigmentation.

In order to understand whether decreased mitochondrial respiration is associated with altered mitochondrial morphology, which is essential to maintain an efficient electron transport chain (ETC) [35], we performed transmission electron microscopy (TEM)-based ultrastructural analysis of mitochondria during melanogenesis (Fig 4D). Our data showed the presence of a few stage III/IV (electron-dense) melanosomes on days 3 and 4, which dramatically increases up to 6 to 8 times on days 5 and 6, respectively (Fig 4E). At the same time, we also observed an increase in the number of unhealthy mitochondria during the late pigmentation phase, as determined by cristae stacking and the presence of vacuoles around mitochondria. The number of unhealthy mitochondria observed is about 40% on day 6, and thus substantially impact mitochondrial respiration (Fig 4F). To determine that the observed mitochondrial changes are attributed to pigmentation and not senescence, we analyze the mitochondrial fragmentation in D6 pigmented cells and PTU-treated D6 depigmented cells using Mitotracker RED dye (S4A Fig). Analysis of individual mitochondria using Image J macro tool for Mitochondrial Network Analysis (MiNA) [36] showed a significantly high round and punctate individual mitochondria in pigmented D6 cells (S4B Fig).

Altogether, from transcriptomic and metabolomics analysis, we propose that melanogenesis can be divided into 3 broad phases—preparatory, melanogenic, and recovery phase. Days 3 and 4 capture the preparatory phase where the MITF-mediated signalling networks are induced and anabolic pathways are activated. Following this, the melanogenic phase on day 5 has heightened pigmentation activity and associated metabolic changes with increased fatty acid utilization. Day 6 profile captures the recovery phase, where pigment inhibitory functions and recovery pathways are up-regulated.

## Epidermal melanogenesis mediator α-MSH activates SREBF1

As fatty acid metabolism is important during pigmentation and the SREBF1 network was among the prominently regulated network in TF-TG analysis, we interrogated the role of SREBF1 during melanogenesis. SREBF1 is an endoplasmic reticulum (ER)-resident protein that is activated upon the cleavage and nuclear translocation of its N-terminal domain. To examine SREBF1 activation, we performed western blot analysis with SREBF1 N-terminal antibody (Fig 5A). Quantitation of cleaved SREBF1 N-terminal band showed increased levels on days 4 and 5, which then slowly declined on day 6 (Fig 5B). Next, we explored whether the classical inducer of melanogenesis, α-MSH, could activate SREBF1. To study this, we designed an activation assay for SREBF1 using a "traffic light construct," wherein the mCherry tag was fused with the N-terminal of SREBF1 and eGFP was fused with C-terminal (S5 Fig). Upon SREBF1 activation, mCherry along with the N-terminal SREBF1 fragment would get translocated to the nucleus while the eGFP remains localized in ER (Fig 5C). Functional validation of this assay was assessed with insulin, which is a known activator of SREBF1 in hepatocytes [37] (S6A Fig). Insulin indeed activated SREBF1 in melanocytes within hours in about 65% of the cells (S6B and S6C Fig). We then carried out live-cell imaging for 4 to 6 hours after α-MSH treatment using confocal microscopy. Representative images show mCherry and eGFP signals in different channels after 4 hours of treatment (Fig 5D). Careful analysis for red signal in the nuclear region shows that the presence of mCherry increases after α-MSH treatment. Time-dependent changes in the colocalization rate of mCherry and Hoechst dye was analyzed after 1

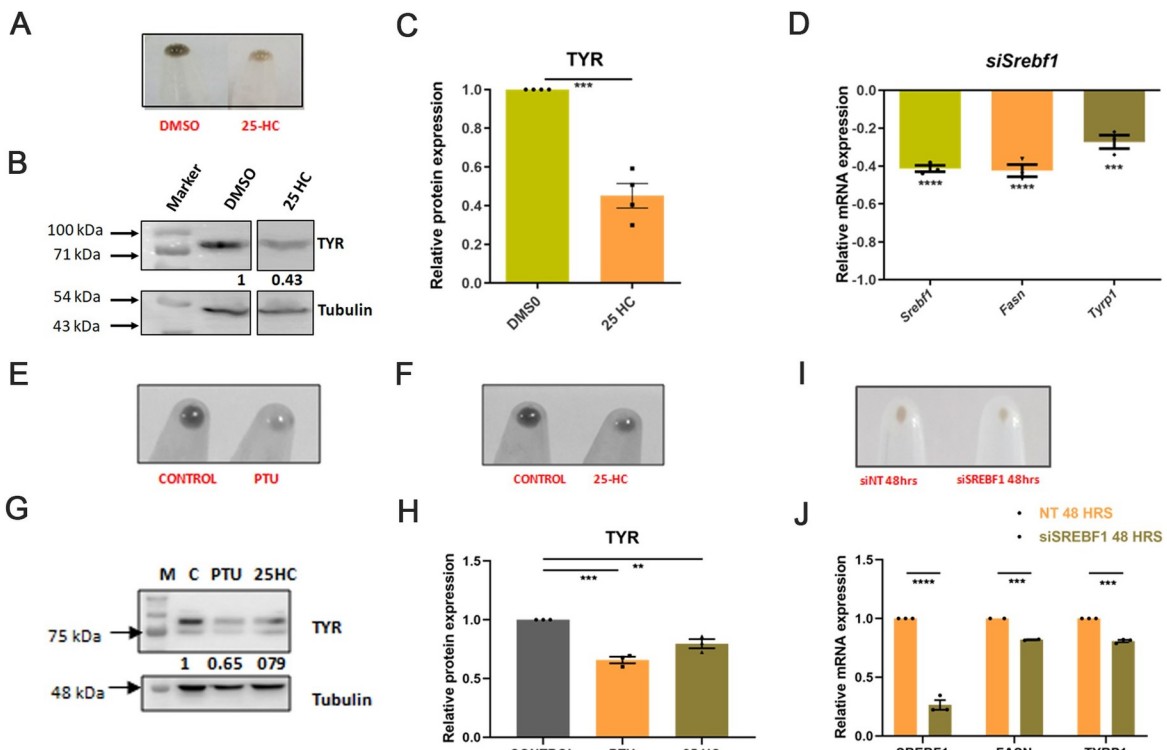

**Fig 6. SREBF1 regulates pigmentation by altering fatty acid metabolism. (A)** Representative B16 cell pellet images showing the phenotypic difference in melanin accumulation on day 6 upon inhibiting SREBF1 activation with 25-HC treatment. **(B)** Representative western blot showing TYR protein expression, normalized to tubulin, upon 25-HC treatment ($N = 4$). Numerical values show average fold change for 4 biological replicates. **(C)** Bar graph depicting quantitative fold change of TYR expression with respect to control. Mean ± SEM is plotted for 4 independent biological replicates. Two-tailed unpaired $t$ test is performed, t = 8.663, df = 6. ***$p$-value = 0.0001. **(D)** Bar graph representing qRT-PCR-based quantitation of *Srebf1*, *Fasn*, and *Tyrp1* genes on D5 upon silencing of *Srebf1* using smart pool siRNA. Mean ± SEM is plotted for 3 biological replicates. One-way ANOVA is applied, F(3,8) = 61.03, $p$-value < 0.0001. Tukey's test is performed for pairwise analysis. ****$p$-Value < 0.001 ***$p$-value < 0.005, **$p$-value < 0.01, *$p$-value < 0.05. **(E)** Representative primary melanocytes cell pellet images showing the phenotypic difference in melanin accumulation upon PTU treatment. **(F)** Representative primary melanocytes cell pellet images showing the phenotypic difference in melanin accumulation upon 25-HC treatment. **(G)** Representative western blot showing TYR protein expression, normalized to tubulin, upon PTU and 25-HC treatment in primary melanocytes ($N = 3$). Numerical values show average fold change for 3 biological replicates. **(H)** Bar graph depicting quantitative fold change of TYR expression with respect to control. Mean ± SEM is plotted for 3 independent biological replicates. One-way ANOVA is applied, F(2,6) = 37.61, $p$-value = 0.0004. Dunnett's multiple comparison test is performed for pairwise analysis. ***$p$-Value < 0.0003, **$p$-value < 0.0038. **(I)** Representative primary melanocytes cell pellet images showing the phenotypic difference in melanin accumulation upon down-regulating *Srebf1* with siRNA. **(J)** Bar graph representing qRT-PCR-based quantitation of *Srebf1*, *Fasn*, and *Tyrp1* genes upon down-regulating of *Srebf1* using smart pool siRNA. Mean ± SEM is plotted for 3 biological replicates. Two-tailed unpaired $t$ test is performed. For *Srebf1* ****$p$-value < 0.0001, for *Fasn* ***$p$-value = 0.0006, for *Tyrp1* ***$p$-value < 0.0001. Quantitative data are provided in S1 Data for Panels C, D, H, and J. PTU, 1-phenyl-2-thiourea; qRT-PCR, quantitative real-time polymerase chain reaction; siRNA, small interfering RNA; 25-HC, 25-hydroxycholesterol.

to 4 hours to follow kinetic activation (Fig 5E). In control cells (DMSO-treated), the colocalization rate of mCherry and Hoechst dye did not increase in a time-dependent manner, suggesting no activation of SREBF1. For α-MSH, we observed that the colocalization rate of mCherry and Hoechst significantly increases in 2 hours. Further, the number of cells showing positive phenotype considerably increases to 60% after α-MSH treatment (Fig 5F). These data suggest that α-MSH induces nuclear translocation of SREBF1 N-terminal in B16 cells. We thus propose the role of the activation of SREBF1 upon pigmentation induction for ensuring efficient fatty acid synthesis.

## SREBF1 regulates pigmentation by altering fatty acid metabolism

After establishing the activation of SREBF1 during pigmentation, we argued whether SREBF1 activation is essential for pigmentation. Towards this, we utilised 25-hydroxycholesterol (25-HC), a pharmacological inhibitor of SREBF1 activation, and evaluated its effect on pigmentation. The inhibitor was added on day 3, and analysis was performed on day 6. Along with the phenotypic change in pigmentation (Fig 6A), quantitative analysis showed a 50% decrease in tyrosinase expression upon 25-HC treatment (Fig 6B and 6C). 25-HC shows broad specificity for both SREBF1 and SREBF2 [38,39]. Thus, we studied the effect of small interfering RNA (siRNA)-based down-regulation of *Srebf1* on pigmentation genes (Fig 6D). Transfections were carried out with Smart pool *Srebf1* siRNA on day 3 and transcriptional changes in pigmentation genes were monitored on day 5. qRT-PCR analysis revealed about 40% to 50% decrease in the *Srebf1* levels (Fig 6D). *Srebf1* down-regulation results in reduced expression of its target *Fasn* on day 5 (Fig 6D). Further, a significant decrease in expression of *Tyrp1* was seen, a gene involved in melanin synthesis, upon *Srebf1* down-regulation (Fig 6D). Next, we examine the effect of PPAR-α/β/γ involvement in melanogenesis. PPARs inhibitors (GW9662 for PPAR-γ, GW6471 for PPAR-α, and GSK3787 for PPAR-β) did not show significant changes in melanin content (S7A Fig). In our RNA-seq analysis, we could only capture expression for PPAR-α and PPAR-γ, but not PPAR-β. Thus, we performed siRNA-mediated down-regulation of PPAR-α and PPAR-γ during pigmentation. Here also, we could not observe a significant effect on pigmentation genes (S7B and S7C Fig). Thus, *Srebf1* could be the major transcriptional regulator of lipid metabolism during pigmentation in the B16 cells.

To further substantiate the effect of pharmacological inhibitor and siRNA-mediated down-regulation of *Srebf1* on pigmentation, we performed similar studies with pigmented primary human melanocytes. We treated primary human melanocytes with 25-HC and PTU for 96 hours and observed a phenotypic decrease in pigmentation in the pellet images of 25-HC and PTU as compared to control (Fig 6E and 6F). These changes are consistent with molecular changes observed in tyrosinase protein expression (Fig 6G and 6H). Next, we studied the effect of siRNA-based down-regulation of *Srebf1* in primary human melanocytes and observed a phenotypic decrease in pigmentation (Fig 6I). We noticed around 70% to 80% decrease in the *Srebf1* levels along with down-regulation of its TG, *Fasn*. Further, *Srebf1* down-regulation induced a significant decrease in the expression of *Tyrp1*, a gene involved in melanin synthesis (Fig 6J). Despite primary melanocytes being pigmented, modest decrease in the expression of pigmentation genes upon *Srebf1* down-regulation indicates that SREBF1 could be a potential target for regulating melanogenesis.

## Fatty acid metabolism is the critical pathway in mediating pigment production

To evaluate the pertinent role of fatty acid metabolism for pigmentation, we performed an inhibitor screen for specific enzymes involved in de novo fatty acid synthesis, TAG synthesis, and lipolysis. We standardised the inhibitor dose using MTT assay and further observed that these standardised doses for different inhibitors do not significantly affect cell count or induce cells death in pigmenting B16 cells (S8A–S8C Fig). B16 cells were treated with FASN inhibitor (C75-20 μM), DGAT inhibitor (T863-10 μM), and lipase inhibitor/FASN inhibitor (Orlistat-50 μM) on day 3 and analysed for pigmentation differences on day 6 (Fig 7A). As expected, C75 and T863 decrease lipid droplet content of the cells on day 5, while Orlistat results in increased accumulation of lipid droplets suggesting that it acts as a lipase inhibitor (S8D Fig). We noted a substantial decrease in the melanin content for all 3 inhibitors, with 60% on Orlistat treatment, 40% on T863 treatment, and 50% on C75 treatment (Fig 7B and

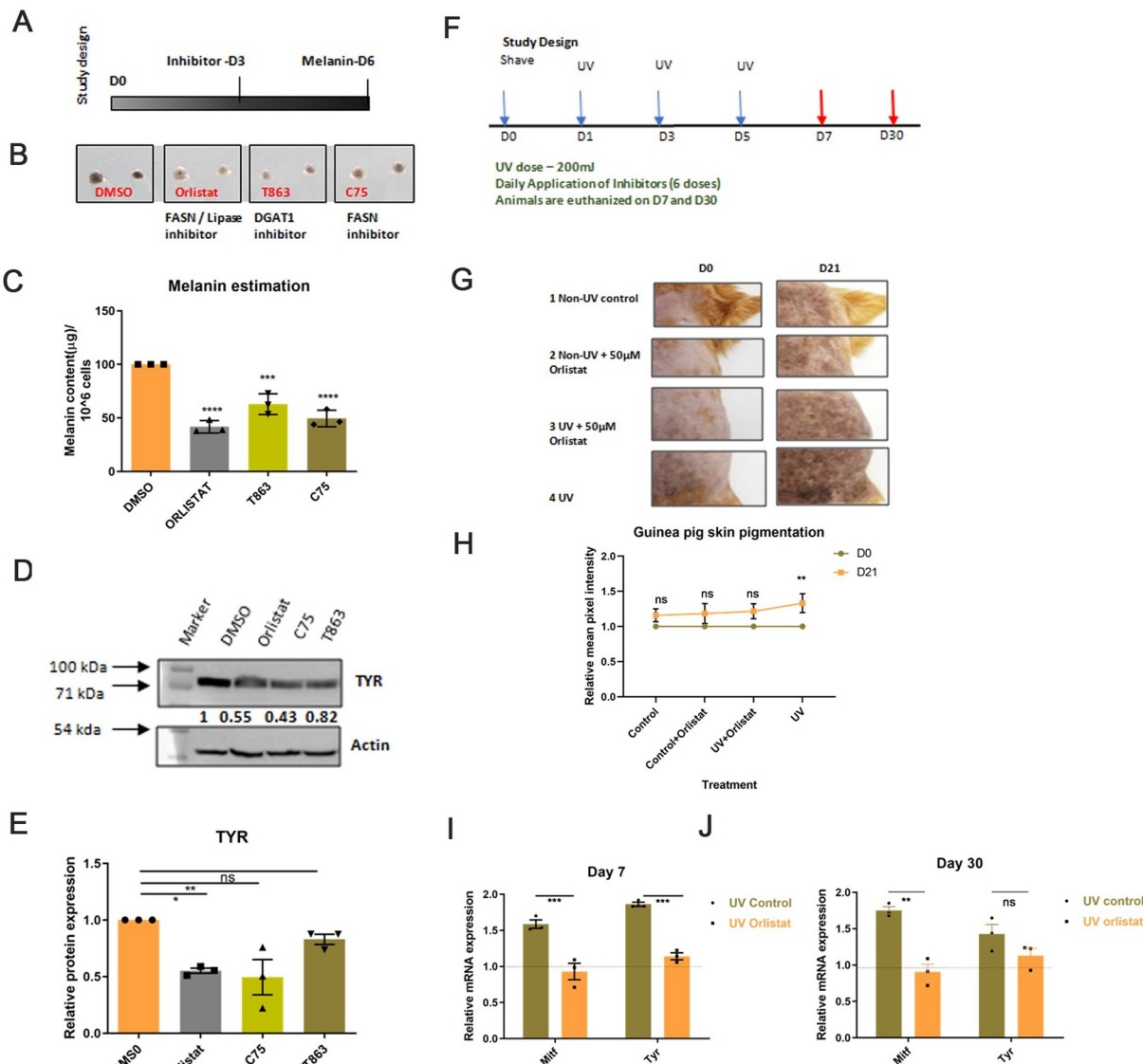

**Fig 7. Fatty acid metabolism is a critical pathway in mediating pigment production.** (**A**) Schematic showing experimental timeline for inhibitor addition and analysis of pigmentation in B16 cells. (**B**) Representative image showing phenotypic differences in melanin accumulation upon inhibitors treatment in 2 technical replicates of D6 cells. Two technical replicates are taken for 3 independent biological sets. (**C**) Bar graph depicting melanin estimation after different inhibitor treatments. Mean ± SEM is plotted for 3 biological replicates. One-way ANOVA is applied, $F(3,8) = 43.13$. $p$-Value < 0.0001. Dunnett's test for pairwise. (**D**) Representative western blot for TYR expression, wrt to Actin, on D6 upon Orlistat, C75 and T863 treatment. Numerical values show average fold change for 3 biological replicates. (**E**) Bar graph depicting quantitative analysis of tyrosinase expression upon Orlistat, C75, and T863 treatment with respect to Actin. Fold change is calculated relative to DMSO. Mean ± SEM is plotted for 3 independent replicates. One-way ANOVA is applied, $F(3,8) = 8.438$, $p$-value = 0.0074. Dunnett's multiple comparison test is performed for pairwise analysis. ***$p$-Value < 0.005, **$p$-value < 0.01, *$p$-value < 0.05. ns is not significant. (**F**) Experimental timeline showing UV-mediated pigmentation induction in guinea pigs. Three doses of UVA+B were given on days 1, 3, and 5 (shown by the blue arrow). Six doses of inhibitor were applied for the first 6 days. Animals were euthanized on D7 and D30 (shown by the red arrow). (**G**) Representative photographs showing phenotypic differences in guinea pig skin pigmentation on D0 and D21. Four segments were marked based on different treatments and are followed temporally on the same animals in the groups of 3. Arrows represent the region showing maximum changes. (**H**) Line plot showing fold change of mean pixel intensity of images on D21 normalized to D0. Fold change was calculated by comparing each segment across days on the same animal. Mean ± SEM is plotted for 3 independent biological replicates. Two-way ANOVA is applied. Sidak's multiple comparison test is performed for pairwise analysis. $p$-Value for Control D0–D21 = 0.1882, $p$-value for Control + Orlistat D0–D21 = 0.1104, $p$-value for UV + Orlistat D0–D21 = 0.0554, $p$-value for Control D0–D21 = 0.0055. (**I**) Bar graph depicting qRT-PCR-based analysis of *Mitf* and *Tyr* expression from epidermal cells on day 7. Mean ± SEM is plotted for 3 individual guinea pigs. One-way ANOVA is applied, $F(3,8) = 36.64$, $p$-value < 0.0001. Tukey's test is performed for pairwise analysis. ***$p$-Value < 0.005, **$p$-value < 0.01, *$p$-value < 0.05. (**J**) Bar graph depicting qRT-PCR-based analysis of *Mitf* and *Tyr* expression from epidermal cells on day 30. Mean ± SEM is plotted for 3 individual guinea pigs. One-way ANOVA is applied, $F(3,8) = 12.87$, $p$-value = 0.002. Tukey's test is performed for pairwise analysis.

***$p$-Value < 0.005, **$p$-value < 0.01, *$p$-value < 0.05. Quantitative data are provided in S1 Data for Panels C, E, H, I, and J. qRT-PCR, quantitative real-time polymerase chain reaction.

7C). The melanin estimation data were recapitulated at the molecular level, as measured by tyrosinase protein levels (Fig 7D and 7E). As inhibition of lipases with Orlistat showed a decrease in pigmentation, we further analyzed the expression of lipase during pigmentation. We observed that the expression of one of the lipases, hormone-sensitive lipase (HSL), increases with pigmentation (S8E Fig). Moreover, a 60% to 70% knockdown of *Hsl* with siRNA showed a significant decrease in *Tyrp1* expression by 25% to 30% (S8F Fig). Altogether, inhibitor data suggest that both fatty acid synthesis and lipolysis arm are important during pigmentation for increasing availability of free fatty acids for further utilization.

To validate the formation of TAGs in pigmented primary human melanocytes, we inhibit DGAT1 enzyme that catalyses the rate-limiting step of diacylglycerol (DAG) to TAG formation using T863 (S8G Fig). To our surprise, pigmented primary human melanocytes have abundant lipid droplets. Treatment with T863 for 8 hours resulted in the complete disappearance of the lipid droplets indicating a high turnover of lipids in pigmented melanocytes (S8H Fig). The above studies highlight the importance of de novo lipogenesis, the storage of lipid droplets and lipolysis as an integral feature of melanogenesis.

Cells can de novo synthesize fatty acid from acetyl CoA or uptake fatty acid from media to accumulate as lipid droplets and utilise for β-oxidation. Thus, we measured the fatty acid uptake ability of cells during pigmentation using C12-Bodipy-labelled fluorescent fatty acids. We observed that in comparison to day 6 cells, days 4 and 5 showed significantly higher uptake (S9A Fig). We further examined the expression of FA transporters in RNA-seq data and found very few of them showing expression values. With qRT-PCR analysis, we observed that *Fabp5*, one of the regulated fatty acid transporter genes in RNA expression analysis showed marginal up-regulation on days 4 and 5 (S9B Fig) and may contribute to fatty acid uptake. Together, our data suggest that both the pathways, FA uptake and de novo fatty acid synthesis are likely to provide fatty acids for β-oxidation. Since fatty acid uptake is dependent on the availability of exogenous fatty acids, our focus was more on de novo fatty acid synthesis.

## Orlistat prevents induction of UV-mediated pigmentation in a guinea pig model

Having established the important role of lipid metabolism in melanogenesis, we were next interested to investigate whether the pharmacological targeting of fatty acid metabolism could show a therapeutic effect in an animal model. We established the guinea pig model to induce skin pigmentation, wherein 200 mJ of UVA and UVB dose were given for 3 alternate days (Fig 7F), as also reported previously [40,41]. Since Orlistat is an FDA-approved drug, we examined whether topical application of Orlistat could be a safe alternative for depigmentation. Orlistat was formulated with PEG8000 and applied topically at 50 μM for 6 days. As expected, we did not observe any toxicological effect of Orlistat on the skin. Anatomically one can observe the subtle level of pigmentation difference in the skin of guinea pigs for different parts of their body. To make our comparison robust, we have performed experiments on the same animals in groups of 3 and followed them temporally. In this study, brown patches on guinea pig skin were divided into 4 sections and subjected to different treatments due to better contrast for hyperpigmentation phenotype than the black patches. The first section corresponds to the non-UV-exposed control region, while the second section is the non-UV-exposed Orlistat treated region. The third section is the UV-exposed region with the application of the Orlistat,

and the fourth section is the UV-exposed control region. Phenotypic changes in the animals were captured on day 21 by taking photographs (Fig 7G). Enhanced induction of pigment was noted in section 4 due to UV exposure. Sections 1 and 2, where the skin is not exposed to UV, show no changes in the pigmentation phenotype. In contrast, the Orlistat-treated UV-exposed region 3 showed reduced pigmentation as compared to the fourth quadrant. Quantitative image analysis suggests a significant increase in pigmentation in the UV-treated section, while the UV + Orlistat–treated section did not show significant changes on day 21 (Fig 7H). These phenotypic changes were confirmed at the molecular level on days 7 and 30 when animals were euthanized. We observed that *Mitf* and *Tyr* expression increased upon UV treatment when compared with non-UV control patch of the skin. While UV Orlistat treatment did not show a similar increase when compared to the non-UV Orlistat skin patch on days 7 and 30 (Fig 7I and 7J). Together, these data suggest that the topical application of Orlistat can be repurposed to decrease hyperpigmentary responses and that modulators of metabolism may be an alternate mechanism to regulate skin pigmentation.

## Discussion

Seminal studies with infiltrating immune cells have underscored the importance of metabolism and metabolites as a guiding force and a critical determinant of the quality and quantity of immune responses [42,43]. However, metabolic rewiring in parenchymal cells like melanocytes to maintain cellular functions is largely unknown. Melanocytes in the human skin epidermis mostly consist of differentiated and non-dividing cells [1]. These cells must rapidly and effectively respond to physiological cues like UV radiations and/or to the secreted factors like cytokines, growth factors, and hormones and regain physiological homeostasis [44,45]. In this study, we have mapped the dynamics associated with transcriptional and metabolic networks that dictate the efficient melanogenesis response of melanocytes. To avoid confounding variables arising from different cell populations, we employed a functionally defined B16 cell pigmentation model that autonomously transits from basal depigmented to the pigmented state over a period of 6 days. This model recapitulates a series of coordinated processes encompassing signalling, transcriptional activation, melanosome biogenesis, melanin synthesis and returns to a homeostatic state. Previous time-series analysis of this pigmentation model had resulted in the identification of interferon-γ signatures in dictating the depigmentation phase of melanogenesis [24]. Our studies here delineate the pivotal role of fatty acid metabolism in melanocyte effector function and also establish the potential therapeutic efficacy of this target.

MITF is the master regulator of melanocyte lineage regulating several genes associated with pigmentation and proliferation [7]. An interesting question is how MITF selectively and differentially activates these genes upon different cues? Some recent studies have provided interesting leads towards this understanding. Malcov-Brog and colleagues showed that MITF expression dynamics tightly control the temporal relationship between the stress response and pigmentation skin protection programs. It is proposed that pigmentation genes have a high affinity for MITF and only small amounts of MITF are required for their expression [26]. Louphrasitthiphol and colleagues suggest that MITF occupancy of promotors is modulated by acetylation [46]. Our group showed enhanced H3K27 histone acetylation at selected differentiation genes facilitate their amplified expression via MITF [47]. In the present study, pigmentation programming is temporally resolved to days (from days 3 to 6), and the above studies could probably explain our observation of different trajectories of protein expression for the classical MITF-mediated pigmentation targets.

Integrated analysis of transcriptomic and metabolomics studies resolved melanocyte pigmentation function into 3 intricately synchronized phases corresponding to preparatory,

melanogenic, and recovery, defined by distinct transcriptional and metabolic signatures. During the preparatory phase, cells start to accumulate precursors and reducing equivalents by increasing the flux of glycolysis towards anabolic pathways like PPP and HBP. The shunting of glycolytic intermediates to the hexosamine biosynthesis module generates substrates for N-glycosylation that will facilitate extensive glycosylation of pigment-producing enzymes, TYR and DCT [8,31,32] and protein components of the melanosome like PMEL17 [10], which are heavily glycosylated in functional form. The flux through PPP remains high during the melanogenic phase that produces NADPH to facilitate the fatty acid biosynthesis crucial in the next phase.

Induction of the melanogenic phase activates fatty acid biosynthesis and accumulation of lipid droplets, which are then rapidly utilised for increased energy requirements of the cell. The onset of melanogenesis triggers gene expression of the central lipid mediator, SREBF1, and its activation maximally around the melanogenic phase. This activation is likely to be triggered by various activators of pigmentation, and we demonstrate α-MSH-mediated cleavage and activation of SREBF1. Increased fatty acid synthesis results in the formation of TAGs. These lipid droplets are much smaller in size than those found in hepatocytes and are rapidly turned over during the continued melanogenic phase through β-oxidation [48]. The accumulation of lipid droplets has been reported earlier within melanocytes in the skin of individuals exposed to therapeutic UV radiations [49]. While lipid metabolism is governed by PPARs and SREBFs, only *Srebf1* knockdown displayed a reproducible modest depigmenting phenotype in melanocytes, in both B16 and primary melanocytes. Although the reasons for this phenotype are unclear, it may be noted that *Srebf1* protein is cleaved and translocated to the nucleus for transcriptional activation, and the transient decrease in its mRNA expression may not be sufficient to completely reverse the pigmentation phenotype. Indeed, the inhibitor 25-HC that targets the activation of SREBF1 shows a better depigmenting effect during the course of treatment. Another possibility is the compensation effect by 2 Srebf1 isoforms, *Srebf1a* and *Srebf1c*. Several inhibitors targeting FASN, DGAT1, and lipase affect pigmentation phenotype, confirming that de novo fatty acid synthesis, storage, and lipolysis are integral in carrying out melanogenesis.

Seahorse-based mitochondrial respiration analysis of these cells during pigmentation show that melanocytes have higher spare respiration capacity for utilizing fatty acids and thus have the potential to make use of this pathway during increased energy requirements. Mitochondrial reliance on fatty acids could result in increased accumulation of free radicals, which is manifested as the accumulation of defective mitochondria in the recovery phase. This corroborates with previous reports suggesting decreased mitochondrial respiration in hyperpigmented cells [17]. The switch to glycolysis and depletion of stored lipid droplets indicates an almost complete reliance on anaerobic glycolysis for the cells' energy needs.

During pathological conditions, it is critical to selectively inhibit the pathways that dysregulate cellular functions such that treatment can restore homeostasis. Since the dynamic nature of metabolic programming among immune cells is linked with their plasticity and function, various inhibitors for metabolic pathways are being perused as a novel therapeutic approach to treat inflammation and autoimmunity [42,50,51]. However, one of the challenges is to generate selectivity during systemic administration for obtaining therapeutic benefits. In our study, by using inhibitors of fatty acid metabolism through a topical formulation, we are able to selectively target melanocyte function to treat hyperpigmentary diseases. We showed that the FDA-approved drug, Orlistat, can be a potential molecule to treat hyperpigmentary responses. In future studies, detailed screening of fatty acid synthesis and lipase inhibitors as depigmenting agents could provide a new class of pigmentation modulators. In conclusion, our study defines principles of cellular homeostasis during melanogenesis that reveals how melanocytes respond

to systemic cues to elicit a physiological response by balancing energetic and cellular stability by sensing the environment. This knowledge can benefit in determining how cutaneous pigmentary diseases develop as well as the means to treat them.

## Material and methods

### B16 pigmentation model and primary melanocyte cultures

B16 melanoma cells were cultured in Dulbecco's modified Eagle's Media (DMEM-high glucose, Sigma-Aldrich, Burlington, Massachusetts, USA, D5648) supplemented with 10% fetal bovine serum (FBS, Gibco, 10270106). Cells were maintained at 5% $CO_2$ levels at 37˚C and grown till 60% to 80% confluency. For the pigmentation model, B16 cells were seeded at a low density of 100 cells/cm$^2$ in DMEM-high glucose media and allowed to gradually pigment for 6 days of the model. Pigmentation was quantitated from pellet images using ImageJ software. As a control, 200 µM PTU (Sigma-Aldrich, P7629) is added on day 2 to maintain depigmented state in the low-density model. B16 cells were treated with C75-20 µM (Sigma-Aldrich, C5490), T863-10 µM (Sigma-Aldrich, SML0539), and Orlistat-50 µM (Sigma-Aldrich, O4139), and 25-HC-500 nM (Sigma-Aldrich, H1015) on day 3 and analyzed for pigmentation differences on day 6.

Adult Human Melanocytes were purchased from Lonza (CC-2586) and cultured in Lonza MGM4 media (CC-3250). These are pigmentated at the basal level. As a control, 200-µM PTU is added to induce depigmentation in these cells. Primary human melanocytes were treated with SREBF1 inhibitor (25-HC-100 nM) for 96 hours.

### UV-induced guinea pig pigmentation model

The 3- to 4-month-old guinea pigs were housed for the experiment. Animals were shaved with depilating cream to expose the skin a day before starting the UV dose. Three doses of 200 mJ of UVA+UVB was given on alternate days to half body of the animal while the remaining half was covered with aluminum foil, which serves as non-UV control skin [40,41]. Both the regions were then applied topically with an equal volume of 50 µM of Orlistat dissolved in 50% of polyethylene glycol 8000 (PEG8000). PEG8000 was used as vehicle control. In each experimental set, a total of 6 doses of drugs were given for the first 6 days. The first drug application was done after first UV dose. Three animals were euthanized at each time point, days 7 and 30. The $1 \times 1$ cm$^2$ of skin was collected from both control and treated region. Epidermis and dermis were separated by treating with 0.25% dispase solution in HBSS for 2 hours at 37˚C. Transcriptional analysis was done for pigmentation genes from RNA isolated from the epidermis. UV-exposed region was compared to the non-UV-exposed region. All animal procedures were performed with an approved protocol (IAEC #521/19) from the Institutional Animal Ethical Committee at the National Institute of Immunology (CPCSEA Registration No- 38/GO/ReBi/SL/99/CPCSEA dated 20.03.17).

### RNA sequencing

RNA isolation was performed using the Triprep RNA isolation kit (Macherey Nagel, Duren Germany, 740966.250) according to the manufacturer's instructions. Briefly, B16 cells were cultured at low density on consecutive days such that days 6 and 3 time points would coincide on the same day. For RNA isolation, an equal number of cells ($5 \times 10^5$) were counted for each time point and stored in the lysis buffer. To obtain high-quality RNA, purification was performed using triprep column. The quality of extracted RNA was assessed by visualizing bands on 1.5% agarose gel and monitoring 260/280 ratio. All the RNA samples were frozen together

**Table 1. List of qRT-PCR primers.**

| Identifier | Gene name | Sequence | Species |
|---|---|---|---|
| RSG 1232 | Gapdh F | AACTGCTTAGCACCCCTGGC | Mouse |
| RSG1233 | Gapdh R | ATGACCTTGCCCACAGCCTT | Mouse |
| IML 359 | Pdk1 F | GGCGGCTTTGTGATTTGTAT | Mouse |
| IML 360 | Pdk1 R | ACCTGAATCGGGGGATAAAC | Mouse |
| IML 335 | Fasn F | CTGCGTGGCTATGATTATGG | Mouse |
| IML 336 | Fasn R | AGGTTGCTGTCGTCTGTAGT | Mouse |
| IML311 | Acaca F | CCTCCGTCAGCTCAGATACA | Mouse |
| IML312 | Acaca R | TTTACTAGGTGCAAGCCAGACA | Mouse |
| IML 313 | Acly F | CCAGTGAACAACAGACCTATGA | Mouse |
| IML 314 | Acly R | AATGCTGCCTCCAATGATG | Mouse |
| RSG8994 | Srebf1 F | GATCAAAGAGGAGCCAGTGC | Mouse |
| RSG8995 | Srebf1 R | TAGATGGTGGCTGCTGAGTG | Mouse |
| RSG 5828 | Tyrp1 F | GATCCGTTCTAGAAGCACCAAGA | Mouse |
| RSG 5829 | Typr1 R | CCTCAGCATAGCGTTGATAGTGA | Mouse |
| RSG 1232 | Gapdh F | AACTGCTTAGCACCCCTGGC | Guinea pig |
| RSG1233 | Gapdh R | ATGACCTTGCCCACAGCCTT | Guinea pig |
| IML 641 | Tyr F | CAGCTTTCAGGCAGAGGTTC | Guinea pig |
| IML 642 | Tyr R | TCCCCAGTATCCAAACTTGC | Guinea pig |
| IML 643 | Mitf F | GAAATTCTGGGCTTGATGGA | Guinea pig |
| IML 644 | Mitf R | ACGCTGTGAGCTCCCTTTTA | Guinea pig |
| IML 1245 | Srebf1 F | AGGTGGAGGACACACTGACC | Human |
| IML 1246 | Srebf1 R | CAGGACAGGCAGAGGAAGAC | Human |
| IML 1247 | Fasn F | AGTACACACCCAAGGCCAAG | Human |
| IML 1248 | Fasn R | GTGGATGATGCTGATGATGG | Human |
| IML 1249 | Tyrp1 | CCGAAACACAGTGGAAGGTT | Human |
| IML 1250 | Tyrp1 | TCTGTGAAGGTGTGCAGGAG | Human |

at −80°C. RNA was precipitated with ethanol to preserve the quality during transportation. A total of 1-μg RNA sample was sent for 2 biological replicates of each time point for RNA sequencing. RNA samples were outsourced to the company, Bencos Research Solutions, for RNA sequencing. RNA with RIN > 8.5 was proceeded for library preparation using TruSeq RNA Sample Prep Kit v2. Sequencing was performed using Illumina NovaSeq 6000 platform. QC check was done using FastQC. Preprocessing to remove trim bases and adaptor was done using Cutadapt. Read alignment was done using HISAT2. GRCm38 was used as the reference genome. Read quantification was performed with HTSeq and normalized counts and differential regulation were obtained using the DESeq2 package in R Studio. Time-course analysis was done using the LRT test, and genes with adjusted $p$-value < 0.001 were taken as significantly differentially expressed from days 3 to 6. Raw counts were transformed using the variance-stabilizing transformation (VST) function, and DEGs were clustered and plotted using the pheatmap package. Genes enriched on the respective days were identified and KEGG pathway enrichment was done using the cluster Profiler package. Selected pathways were plotted using ggplot2 in R Studio. DEGs were analysed for the TG enrichment of TF using the TRRUST database on metascape (metascape.org). Top 7 TFs from each day (except Trp53 as it was enriched on all the days) were plotted as a TF-TG network using cytoscape. The colour gradient represents the scaled (row-wise) expression values from the RNA Seq data.

## cDNA synthesis and real-time PCR

RNA was reverse transcribed using the Superscript III cDNA synthesis kit (Life Technologies, Carlsbad, California, USA, 18080–051) according to the manufacturer's protocol. Gene expression analysis by quantitative real-time PCR was performed on a Roche Light Cycler 480 II real-time cycler using the SYBR GREEN qPCR Master Mix (Kapa Biosystems, Massachusetts, USA, KM4101) to evaluate transcriptional regulations. Most of the primers were designed using Primer3 and checked by the NCBI Primer blast tool. List of primers used for qRT-PCR is provided in the Table 1. Gene-specific primers were obtained from Sigma Aldrich. Either *Hgprt or Gapdh* was used as the normalizing control and quantification was done by the comparative Ct method.

## Cell lysate preparation and western blotting

B16 melanoma cells were washed with ice-cold Dulbecco's phosphate buffer saline (DPBS-Gibco, Thermo Fisher Scientific, Waltham, Massachusetts, USA, 14190144) and lysed with NP40 cell lysis buffer (Invitrogen, Waltham, Massachusetts, USA, FNN0021) supplemented with protease inhibitor cocktail (Roche, Basel Switzerland, 04693132001). Cells were incubated with NP40 for 30 minutes on ice. The soluble fractions of cell lysates were isolated by centrifugation at 13,000 rpm in a refrigerated microcentrifuge for 30 minutes. The protein concentration in the soluble fraction was quantified using the bicinchoninic acid (BCA) protein estimation kit (Thermo Fisher Scientific, Waltham, Massachusetts, USA, 23227). Known concentrations of bovine serum albumin (BSA) was used to plot the standard curve. The 30 to 50 μg of the protein was boiled in SDS dye and separated on 10% SDS PAGE gel. Tyrosinase antibody is synthesized from Genescript. PMEL17 (ab137078), MITF (ab12039), and FASN (ab22759) antibodies are procured from Abcam, while SREBF1 (04–469) is ordered from Millipore. HRP-conjugated Actin (ab8227) and Tubulin (ab6046) are used as a loading control. Horseradish peroxidase-conjugate Anti-Mouse (NA931) and Anti-Rabbit (NA934) antibodies are obtained from GE Healthcare. For western blot standard-enhanced chemiluminescence reagents (WBLUF0100) were used from Millipore. ImageJ software was used for quantification.

## Polar metabolite extraction for mass spectrometry and NMR

For liquid chromatography-coupled tandem mass spectrometric analysis, polar metabolite extraction was performed using 80% Methanol-Water solvent (LC-MS grade Methanol-Merck 106035, LC-MS grade Water-Fluka 39252). Cells were grown in multiple flasks to obtain $10^6$ cells per replicate. The 3 to 4 replicates were prepared for each time point. B16 cells were washed with cold DPBS and trypsinized with 0.1% trypsin (Gibco, 15090046 diluted in Versene, Gibco, 15040066). Cells were harvested in defined trypsin inhibitor (Gibco, R007100), and 1 million cells were counted for each sample and proceeded for metabolite extraction. Again, cells were washed with DPBS and pelleted down at 500 g for 5 minutes at 4°C. Approximately 500 μL of chilled 80% methanol was added to the cell pellet. For extraction, samples were mixed well by vortexing and incubated on dry ice for 10 minutes. Penicillin (Gibco, 15140122) was added in 80% methanol as an internal control during extraction. Samples were spun at 13,000 rcf for 30 minutes at 4°C. The supernatant was collected in a chilled 1.5 ml microcentrifuge tube and samples were dried in a SpeedVac vacuum concentrator. Samples were further lyophilized and stored at −80°C till mass spectrometric run. The samples were run on a Waters Xevo-TQS tandem mass spectrometer coupled to Acquity UPLC. The analysis was performed using MassLynx software, followed by analysis with MetaboAnalyst 5.0 for pathway enrichment. Fold changes were plotted for each metabolite.

Pulse-chase labeling of glycolysis and TCA metabolites was performed using NMR. The 10 mM [U-$^{13}$C]-Glucose (Cambridge Isotope Laboratory, Andover and Tewksbury,

Massachusetts, USA, CLM-1396-10) was fed in glucose-free RPMI media (Himedia, Mumbai, Maharashtra India, AT150) for 4 hours at each time point. Due to the low sensitivity of NMR, extraction was done from 20 million cells per sample. The 200 μM [U-$^{13}$C]-palmitate (Cambridge Isotope Laboratory, CLM-409-0.5) was added for 24 hours in glucose-free RPMI at each time point. Metabolite extraction was done with the same protocol as mention above. Samples were dissolved in 160 μL of deuterated water ($D_2O$) and transferred in 3 mm NMR tubes. All NMR measurements were performed on a 500 MHz Bruker Avance III spectrometer equipped with 5-mm TCI cryo-probe. Topspin 3.6 pl7 (Bruker) was used for data acquisition, Fourier transformation and processing of data. Two-dimensional [$^{13}$C,$^1$H] heteronuclear single quantum coherence [HSQC] experiments were measured at 310 K. The 2D [$^{13}$C,$^1$H] HSQC, spectra were measured with a spectral width of 7002.8 Hz along the $^1$H dimension, and 22639.57 Hz along the $^{13}$C dimension. A total of 16 dummy scans and 96 scans with a relaxation delay of 1.5 second was used for a total acquisition time of 146 ms ($t_{2max}$) along $^1$H dimension and 2.8 ms ($t_{1max}$) along $^{13}$C dimension. Processing was performed using 90˚ shifted sine-square bell window function for both dimensions. Peak correlation and peak intensity calculation were performed using Computer Aided Resonance Assignment (CARA) software [52]. Chemical shift values were assigned to specific metabolites using the Biological Magnetic Resonance Bank (BMRB) (http://www.bmrb.wisc.edu) and Human Metabolome Database (HMDB) (http://www.hmdb.ca). A chemical shift error tolerance of 0.05 ppm and 0.5 ppm was used for $^1$H and $^{13}$C chemical shifts, respectively. Fold change as compared with day 0 was plotted for each metabolite.

## Glucose measurement

Glucose concentration in media was measured on COBAS INTEGRA 400 using a glucose detection kit. It is based on the principle of an enzymatic assay where glucose is first converted to glucose-6-phosphate by hexokinase, followed by oxidation of glucose-6-phosphate by G6PD, which is coupled to the reduction of NAD to NADH. NADH produced in the reaction reduces the colourless probe to a coloured product with strong absorbance at 450 nm. The amount of glucose is equivalent to NADH produced. A standard curve was generated using glucose standards ranging in between 0 to 10 g/L concentration. The instrument is highly accurate up to the range of 6 g/L. The 200 μl residual media was collected on each day from days 3 to 6 in specialized tubes and placed in the COBAS instrument. Reagents were mixed automatically and readings were recorded. Glucose concentration in the media on different days was calculated from the standard curve.

## Lactate measurement

Lactate was measured in cell lysate with the Sigma Lactate assay kit (MAK064) using the manufacturer's protocol. Briefly, an equal number of cells ($10^6$) were harvested on days 5 and 6 and the cellular lysate was prepared using the given lysis buffer. Lactate standards were prepared in a range of 0 to 10 nmole in 50 μL of assay buffer. An equal volume of a master mix containing lactate assay buffer, lactate enzyme, and lactate probe were added to standards and samples. The plate was incubated for 30 minutes at room temperature and colorimetric measurement was recorded at 570 nm.

## Mitochondrial oxygen consumption rate measurement

Oxygen consumption rate was measured using the Seahorse Mito Stress Kit (Agilent, Santa Clara, California, USA, 103015–100) according to the manufacturer's protocol. Briefly, B16 cells were grown in low density in DMEM high glucose media. The 3 separate cultures were

setup on consecutive days so as to terminate all the time points (D5 and D6) on the same day. One day before termination of the cultures, cells were trypsinized and 60,000 cells were seeded in a seahorse culture plate and allowed to adhere overnight. Cell density was first standardized to get reading in an appropriate range. Seahorse media was freshly reconstituted by adding glutamine (2 mM), sodium pyruvate (1 mM), and glucose (10 mM) or oleate (200 μM) and equilibrated to 37˚C for each experiment. pH was set to 7.4 by adding NaOH solution. Once the media was prepared, cells were washed twice with DPBS. A total of 500 μl of freshly pre-pared media was added to each well, from the wall. Cells were incubated in Seahorse media in a non-$CO_2$ incubator for 1 hour before the assay. Inhibitors were added as per the kit direction in different ports just before starting the assay. Assay run for 2 hours and data generated was analyzed using WAVE software.

## Immunofluorescence measurements

Lipid droplets were measured using BODIPY 493/503 dye (Invitrogen, D3922). B16 cells were grown on 1 $cm^2$ coverslip in low density (1,000 cells per coverslip) for different days. Cells were fixed by adding 4% methanol-free formaldehyde (Thermo Fischer Scientific, 28906) on the coverslip and incubated at room temperature for 20 minutes. 3X DPBS washes were given to remove 4% methanol-free formaldehyde. Cells were stored in DPBS at 4˚C till the last time point. Fixed cells were incubated with 10 μM BODIPY dye in DPBS for 1 hour at room tem-perature. 3X DPBS washes were done to remove excess stain. Coverslip was mounted with Gold antifade DAPI solution (Thermo Fischer Scientific, P36931). Imaging was done on the Leica SP8 confocal microscope. Three biological replicates were imaged for each experiment, with 30 cells per set in the B16 model.

Mitochondrial morphology was measured using Mito Tracker Red dye. B16 cells were cul-tured in 2-well live imaging chamber dishes with an area of 1 $cm^2$ in low density (500 cells per coverslip) for different days. Live cells were incubated with 500 nM Mito Tracker Red dye for 30 minutes at 37˚C in DMEM without FBS. Three DPBS washes were given. Imaging was done on the Leica SP8 confocal microscope. Three biological replicates were imaged for each experiment, with 100 cells per set.

B16 cells overexpressing pmCherry-Srebf1-eGFP (traffic light construct) were seeded in live imaging chambers (2 chambered live imaging chambers from Nunc). The nucleus was stained with Hoechst 33342 solution (Thermo Fischer Scientific, 62249). Approximately 3.5 μM Insulin (Sigma-Aldrich, I6634) or 10 μM α-MSH (Sigma-Aldrich, M4135) was added just before setting up the live imaging module in the Leica SP8 Confocal microscope. Trans-fected cells were selected at random using the "Mark & Find" feature, and 30 to 40 cells were imaged for a duration of 4 to 6 hours per replicate. The experiment was performed in triplicates.

## siRNA transfection

siRNA transfections were performed in T75 flasks on day 3 of the pigmentation model. A total of 100 nM of siRNA was added per flask with a 1: 3 times V: V ratio of Dharmafect transfection reagent. siRNA was commercially procured from Qiagen (FlexiTube GeneSolution GS20787 for Mouse Srebf1, FlexiTube Gene Solution 1027281 for negative C siRNA). Other siRNAs were purchased for Dharmacon (ON-TARGETplus Mouse Lipe 16890 siRNA Smart pool for HSL, ON-TARGETplus Human SREBF1 (6720) siRNA Smart pool). The transfection was done in opti-MEM (Gibco 31985070) media for 6 hours. Post transfection, the opti-MEM media was removed and cells were washed with 1X DPBS and then the day 3 culture media was added back to the cells. siRNA transfection in primary melanocytes was done using

**Table 2. List of cloning primers.**

| Identifier | Gene name | Sequence | Species |
|---|---|---|---|
| IML524 | Srebf1c F XhoI | CTCGAGggatggattgcacatttgaa | Mouse |
| IML581 | Srebf1c middle R | Cgggccagagtgtggcctagt | Mouse |
| IML580 | Srebf1c middle F | Gccaatggactactagtgttg | Mouse |
| IML587 | Srebf1c R EcoRI | GAATTCgctggaagtgacggtggt | Mouse |

Nucleofection Kit. Media is changed after 24 hours of transfection. Cells were harvested at 48 hours time point to capture transcriptional changes.

## Melanin estimation

Melanin estimation was performed as described earlier [53]. Briefly, cells were lysed in 1 N NaOH by heating at 80˚C for 2 hours, and then, absorbance was measured at 405 nm. Finally, the melanin content was estimated by interpolating the sample readings on the melanin standard curve obtained with synthetic melanin.

## Live/Dead cell analysis

Live dead staining is performed using Propidium Iodide dye (493/636). Briefly, the cells were trypsinised and washed with DPBS and incubated with 2.5 μg/ml Propidium Iodide Dye at room temperature for 1 minute. As a positive control, cell death was induced by incubating cells at 55˚C for 20 minutes. Samples were run on flow-cytometry. The percentage of PI-positive cells were plotted.

## Fatty acid uptake assay

Fatty acid uptake assay was performed using BODIPY FL C12 Dye (505/511) (Invitrogen, D3822). Briefly, the cells were trypsinised, washed with DPBS and incubated with 1 μM BODIPY FL C12 Dye at 37˚C for 30 minutes. Subsequently, cells were washed thrice with DPBS and run-on flow-cytometry. Medium fluorescent intensity was analysed for different days.

## Cloning

To clone *Srebf1*, we amplified the common splice variant, *Srebf1c*, using PCR. List of primers used to amplify *Srebf1* is provided in Table 2. To get a complete 3.4 kb fragment, we amplified the first 1.6 kb separately and the last 1.7 kb sequence separately. The complete sequence was amplified with overlapping PCR by using 2 amplicons specific to the first and second half sequence of *Srebf1c*. The amplicon was phosphorylated with T4 polynucleotide kinase. The plasmid pBluescript (pBS-SK+) was digested with EcoRV and end were dephosphorylated with calf intestinal phosphatase (CIP). Amplified full-length sequence of *Srebf1c* was cloned in pBS-SK+ vector in EcoRV restriction digestion site. From Srebf1-pBS plasmid, *Srebf1c* fragment was removed by doing partial restriction digestion with XhoI and EcoRI enzymes. Complete digestion gives 2 fragments, from 1 to 493 bp and 494 to 3,410 bp, while partial digestion gives 3 bands corresponding full length 1 to 3,410 and 1 to 493 bp and 494 to 3,410 bp. We extracted the full-length fragment from agarose gel corresponding to 3,335 bp and subcloned it in the mCherry-C1 vector. In our lab, eGPF was already cloned in the mCherry-C1 vector and stored in lab repository as clone number pAK4.0. With XhoI and EcoRI restriction enzymes, we subcloned *Srebf1c* in mCherry-eGFP vector. NEB enzymes and reagents are used for cloning.

## Transmission electron microscopy

Cells were fixed in fixative containing 2.5% glutaraldehyde and 4% paraformaldehyde in 0.1 M sodium cacodylate buffer (pH 7.2) at room temperature for 4 hours and then rinsed in PBS. Fixed cells were embedded in 2% agar and post-fixed in 1% osmium tetraoxide for 1 hour. Samples were dehydrated in graded series of ethanol (50% to 100%) and then subjected to propylene-oxide for 30 minutes and infiltrated with increasing proportions of propylene-oxide: Epon (2:1, 1:1 and 1:2) for 2 hours and embedded in Epon resin and polymerized for 72 hours at 60˚C. Ultrathin sections (63 nm) were cut on Leica Ultra-microtome, placed on copper grids and stained with uranyl acetate and lead citrate, and examined on a 200-KV Tecnai G2 Twin transmission electron microscope (FEI make).

## Statistical analysis

All statistical analysis was performed using GraphPad Prism 8 software. Unpaired Student $t$ test, one-way ANOVA, or two-way ANOVA was applied to compare the significant difference between the means of 2 or multiple groups respectively. Tukey's test was applied to compare the pairwise sample. Dunnett's test was applied to compare with the control group. The $p$-values obtained during analysis are indicated in the figure legends.

## Supporting information

**S1 Fig. Validation of RNA sequencing data.** (**A**) PCA plot from data obtained from unbiased transcriptomic analysis from 2 independent biological replicates. PC1 is 80.33% and PC2 is 13.81%. (**B**) Correlation plot showing correlation value R obtained between 2 biological replicates in RNA sequencing data. R ranges from 0.99–0.1, with high $p$-value 2.2e-16. (**C**) Heatmap representing the expression of fatty acid synthesis genes from days 3 to 6 in RNA sequencing data. Scale from blue to red represents z-score for normalized expression values from –1 to +1. (**D**) Heatmap representing the expression of TAG synthesis genes from days 3 to 6 in RNA sequencing data. Scale from blue to red represents z-score for normalized expression values from –1 to +1. (**E**) Representative image showing pigmented cell pellet in the low-density B16 pigmentation model for days 3 to 6 vs. PTU-treated depigmented cells. (**F**) Bar graph depicting cell count from days 4 to 6 in control and PTU-treated low-density cells in 3 independent biological sets. One-way ANOVA is applied, $F_{(5,12)} = 203.3$. Turkey's test is performed for pairwise comparison. *$p$-Value $< 0.0142$. (**G**) Representative western blot for FASN expression on D3–D6 during melanogenesis with respect to Actin. Numerical values show average fold change for 3 biological replicates. (**H**) Bar graph depicting quantitative analysis of FASN during melanogenesis for 3 independent biological replicates. Mean ± SEM is plotted for 3 independent biological replicates. One-way ANOVA is applied $F_{(3,7)} = 25.74$, $p$-value $= 0.0004$. (**I**) Representative western blot for FASN expression on D4–D6 PTU-treated cell-seeded at low density with respect to Actin. Numerical values show average fold change for 3 biological replicates. (**J**) Bar graph depicting quantitative analysis of FASN during melanogenesis for 3 independent biological replicates. One-way ANOVA is applied $F_{(2,6)} = 5.288$, $p$-value $= 0.0474$. Quantitative data are provided in S2 Data for Panels A, B, F, H, and J. PCA, principal component analysis; PTU, 1-phenyl-2-thiourea; TAG, triacylglycerols. (TIF)

**S2 Fig. Validation of transcriptional network analysis.** (**A**) Bar graph depicting qRT-PCR of *Mitf* with respect to *Hgprt*. Mean ± SEM is plotted for 3 independent biological replicates. One-way ANOVA is applied. For *Mitf*, $F_{(3,8)} = 16.59$, $p$-value $= 0.0009$. Turkey's test is performed for pairwise comparison. (**B**) Bar graph depicting qRT-PCR of *Tyr* with respect to

*Hgprt*. Mean ± SEM is plotted for 3 independent biological replicates. One-way ANOVA is applied. For *Tyr*, F (3,8) = 24.08, *p*-value = 0.0002. Turkey's test is performed for pairwise comparison. (**C**) Bar graph depicting qRT-PCR of *Srebf1* with respect to *Hgprt*. One-way ANOVA is applied. For*Srebf1*, F (3,8) = 21.72, *p*-value = 0.0003. Turkey's test is performed for pairwise comparison. (**D**) Bar graph depicting qRT-PCR-based analysis fatty acid synthesis genes, *Fasn*, *Acaca*, *Acacb*, and *Acly*, with respect to *Hgprt*. Mean ± SEM is plotted for 3 independent biological replicates. One-way ANOVA is applied separately for each gene. For *Fasn* F (3,8) = 8.435, *p*-value = 0.0074, for *Acaca* F (3,8) = 9.811, *p*-value = 0.0063, for *Acacb* F (3,8) = 10.72, *p*-value = 0.0035, for *Acly* F (3,8) = 11.06, *p*-value = 0.0032. (**E**) Bar graph depicting qRT-PCR of *Nrf2* with respect to *Gapdh*. Mean ± SEM is plotted for 3 independent biological replicates. One-way ANOVA is applied. For *Nrf2*, F (3,8) = 9.622, *p*-value = 0.005. Turkey's test is performed for pairwise comparison. (**F**) Bar graph depicting qRT-PCR-based analysis *Nrf2* TGs, *Gsr* and *Gst*, with respect to *Gapdh*. Mean ± SEM is plotted for 3 independent biological replicates. One-way ANOVA is applied separately for each gene. For *Gst*, F (3,7) = 7.68, *p*-value = 0.0128, for *Gsr*, F (3,8) = 9.666, *p*-value = 0.0049. Quantitative data are provided in S2 Data for Panels A–F. qRT-PCR, quantitative real-time polymerase chain reaction; TG, target gene.
(TIF)

**S3 Fig. Temporal resolution of metabolic signature of melanocytes during pigmentation.** (**A**) PCA plot depicting segregation of different days based on metabolite signatures. Groups 1 to 4 correspond to days 3 to 6. Four biological replicates are taken for each time point. PC1 is 48.9% and PC2 is 18.5%. (**B**) Heatmap depicting top 50 regulated metabolites across different days for each replicate. Blue colour arrows are used to show amino acids and nucleotides, higher on D3 and D4, while orange arrows are shown to mark cofactors and TCA metabolites, higher on D5 and D6. (**C**) Bar graphs depicting glucose utilization upon pigmentation induction in B16 cells cultured at low-density model. Glucose utilization is calculated by subtracting glucose in media on consecutive days (i.e., glucose concentration on (n-1) day-glucose concentration on nth day)/cell number on nth day*100. Data are represented for 3 independent replicates. Mean ± SEM is plotted for 3 biological replicates. One-way ANOVA is applied F(3,7) = 39.50, ****p*-value < 0.0001. (**D**) Schematic showing biochemical reactions and enzymes involved in conversion of pyruvate to lactate and acetyl CoA formation. (**E**) Bar graph depicting quantitation of amount of lactate in cellular lysate on D5 and D6 for 3 independent replicates. Mean ± SEM is plotted for 3 biological replicates. Two-tailed Student *t* test is applied, t = 3.284, df = 4, *p*-value = 0.0304. (**F**) Bar graph depicting qRT-PCR-based mRNA expression of *Pdk1*, which regulates PDH activity on D5 and D6 for 3 independent replicates.
Mean ± SEM is plotted for 3 biological replicates. Two-tailed Student *t* test is applied,
t = 3.541, df = 4, *p*-value = 0.0240. Quantitative data are provided in S2 Data for Panels C, E, and F. PCA, principal component analysis; qRT-PCR, quantitative real-time polymerase chain reaction; TCA, tricarboxylic acid cycle.
(TIF)

**S4 Fig. Mitochondrial fragmentation analysis in pigmented and depigmented cells. (A)** Representative confocal microscopy images showing mitochondrial morphology in pigmented day 6 cells vs. PTU-treated depigmented cells on day 6 using MitoTracker RED dye. Images were taken at 63×. Scale is 5 μm. (**B**) Bar graph depicting the quantitation of fragmented mitochondria in pigmented vs. PTU-treated depigmented day 6 cells using ImageJ macro tool MiNA. Approximately 100 cells were taken in each replicate. Mean ± SEM is plotted in 3 independent biological replicates. Mean ± SEM is plotted for 3 biological replicates. Two-tailed Student *t* test is applied, t = 2.262, df = 24, *p*-value = 0.0330. Quantitative data are provided in

S2 Data for Panel B. MiNA, Mitochondrial Network Analysis; PTU, 1-phenyl-2-thiourea.
(TIF)

**S5 Fig. Cloning of *Srebf1* in mCherry-C1 eGFP construct. (A)** Representative agarose gel showing amplification of first and second half of *Srebf1* gene. (**B**) Representative agarose gel showing amplification of full-length *Srebf1* using overlap PCR. (**C**) Representative agarose gel showing partial digestion of Srebf1-pBS clone, Srebf1 to obtain full-length fragment for subcloning in mCherry-C1 eGFP vector. (**D**) Expected size of *Srebf1* pBS clone is tabulated upon digestion with XhoI and EcoRI. (**E**) Representative agarose gel showing clone confirmation of mCherry-Srebf1-eGFP vector. (**F**) Expected size of mCherry-Srebf1-eGFP vector is tabulated upon restriction digestion. (**G**) Representative western blot for expression analysis of mCherry-Srebf1-eGFP construct upon transfection in B16 cells.
(TIF)

**S6 Fig. Standardization of SREBF1 activation assay. (A)** Representative fluorescence images showing eGFP, mCherry, and merge signal. Magnified nuclear images were shown to focus on SREBF1 nuclear translocation after 4 hours of insulin treatment. Around 20 cells were analyzed in each of the 3 biological replicates. (**B**) Dot plot depicting colocalization rate between mCherry and Hoechst signal analyzed for each cell at 1 to 4 hours after insulin treatment. Mean ± SEM is plotted for 3 biological replicates. One-way ANOVA is applied, $F(2.516, 52.83)$ = 8.234, $p$-value = 0.0003. Dunnett's multiple comparison test is performed. ****$p$-Value < 0.0001, ***$p$-value = 0.00081, **$p$-value = 0.0011. (**C**) Bar graph depicting quantitation of number of cells showing positive phenotype after Insulin treatment, determined by increased colocalization rate of mCherry and Hoechst signal from 1 to 4 hours. Mean ± SEM is plotted for combined analysis. Two-tailed Student $t$ test is performed, t = 5.131, df = 4, **$p$-value = 0.0068. Quantitative data are provided in S2 Data for Panels B and C.
(TIF)

**S7 Fig. PPAR signalling do not regulate pigmentation in the B16 model. (A)** Representative B16 cell pellet images showing melanin accumulation phenotype on day 6 upon inhibiting PPAR-γ with GW9662, PPAR-α with GW6471, and PPAR-β with GSK3787 ($N$ = 3). Bar graph depicting melanin estimation after different inhibitor treatments. Mean ± SEM is plotted for 3 biological replicates. One-way ANOVA is applied, $F(3,8)$ = 1.728. $p$-Value = 0.2383. Tukey's test is performed for pairwise analysis. ns is nonsignificant. (**B**) Bar graph representing qRT-PCR-based quantitation of *Ppara* and *Tyrp1* genes on D5 upon silencing of *Ppara* using smart pool siRNA. Mean ± SEM is plotted for 3 biological replicates One-way ANOVA is applied, $F(3,6)$ = 49.29. For NT vs. *Ppara*, ***$p$-value = 0.0002. For NT vs. *Tyrp1*, $p$-value is nonsignificant. (**C**) Bar graph representing qRT-PCR-based quantitation of *Pparg* and *Tyrp1* genes on D5 upon silencing of *Pparg* using smart pool siRNA. Mean ± SEM is plotted for 3 biological replicates. One-way ANOVA is applied, $F(3,6)$ = 1.188. For NT vs. *Ppara*, *$p$-value = 0.0255. For NT vs. *Tyrp1*, $p$-value is nonsignificant. Quantitative data are provided in S2 Data for Panels A–C. qRT-PCR, quantitative real-time polymerase chain reaction; siRNA, small interfering RNA.
(TIF)

**S8 Fig. Effect of lipid metabolism perturbation on melanogenesis. (A)** Bar graph represents inhibitor dose standardization in B16 cells using MTT assay. Red arrows represent the dose chosen for further experiments. (**B**) Bar graph representing cell count on day 6 upon treatment with Orlistat, C75, T863, and PTU. Mean ± range is plotted for 2 biological replicates. ns is nonsignificant. One-way ANOVA is applied $F(4,5)$ = 3.964. $p$-Value = 0.0922. (**C**) Bar graph depicts cell death induced in day 6 cells upon inhibitor treatment measured using Propidium

Iodide staining. Mean ± SEM is plotted for 3 biological replicates. (**D**) Representative confocal microscopy images showing lipid droplet accumulation in B16 cells during pigmentation upon addition of treatments (Orlistat, C75, T863) on D3 and analysis on D5. Images were taken at 63×. Scale is 5 μm. (**E**) Bar graph depicting qRT-PCR of *Hsl* with respect to *Hgprt*. Mean ± SEM is plotted for 3 independent biological replicates. One-way ANOVA is applied. For *Hsl*, F (3,8) = 15.92, *p*-value = 0.001. Turkey's test is performed for pairwise comparison. (**F**) Bar graph representing qRT-PCR-based quantitation of *Hsl* and *Tyrp1* genes on D6 upon silencing of *Hsl* using smart pool siRNA. Mean ± SEM is plotted for 3 biological replicates. One-way ANOVA is applied, F(3,4) = 61, ***p*-value = 0.0009. Turkey's test is performed for pairwise comparison. (**G**) Schematic study design for analyzing TAG formation in primary human melanocyte culture by inhibiting DGAT1 using T863, and capturing lipid droplets content in these cells using BODIPY dye. (**H**) Representative confocal microscopy images showing lipid droplet accumulation in primary melanocytes upon T863 addition. Images were taken at 63×. Scale is 5 μm. Quantitative data are provided in S2 Data for Panels A, B, C, E, and F. PTU, 1-phenyl-2-thiourea; qRT-PCR, quantitative real-time polymerase chain reaction; siRNA, small interfering RNA; TAG, triacylglycerol.
(TIF)

**S9 Fig. Role of fatty acid uptake during pigmentation.** (**A**) Representative plot depicts median fluorescent intensity corresponding to the uptake of C-12 fluorescently labelled fatty acid from days 4 to 6. Mean ± SEM is plotted for 3 biological replicates. One-way ANOVA is applied, F(2,6) = 72.66. Turkey's test is performed for pairwise comparison. For D4 vs. D5, *p*-value is nonsignificant. For D4 vs. D6, ****p*-value < 0.0001. (**B**) Bar graph depicting qRT-PCR of *Fabp5* with respect to *Hgprt*. Mean ± SEM is plotted for 3 independent biological replicates. One-way ANOVA is applied. Turkey's test is performed for pairwise comparison. For *Fabp5*, F (3,8) = 5.374, *p*-value = 0.0255. ns is nonsignificant. Quantitative data are provided in S2 Data for Panels A and B. qRT-PCR, quantitative real-time polymerase chain reaction.
(TIF)

**S1 Data. Data underlying panel Figs 1A, 1C, 1G, 1H, 3C, 3E, 4A, 4B, 4C, 4E, 4F, 5B, 5E, 5F, 6C, 6D, 6H, 6J, 7C, 7E, 7H, 7I, and 7J.**
(XLSX)

**S2 Data. Data underlying panel Supporting information S1A, S1B, S1F, S1H, S1J, S2A–S2F, S3A–S3C, S3E, S3F, S4B, S6B, S6C, S7A–S7C, S8A–S8C, S8E, S8F, S9A, and S9B Figs.**
(XLSX)

**S3 Data. Data underlying panel Figs 1E, 1D, and 2.**
(XLSX)

**S4 Data. Data underlying panel Fig 3A.**
(XLSX)

**S1 Raw Images. Raw data underlying panel Figs 1B, 5A, 5D, 6B, 6G, 7D, S1G, S1I, and S6A.**
(PDF)

# Acknowledgments

We acknowledge the infrastructure supported by CSIR-IGIB for imaging facility and Mr. Manish for help with imaging. We thank Department of Biotechnology, Government of India

and ICGEB, New Delhi core research fund for providing financial support for the high field NMR spectrometer at ICGEB, New Delhi.

## Author Contributions

**Conceptualization:** Farina Sultan, Vivek T. Natarajan, Rajesh S. Gokhale.

**Data curation:** Farina Sultan.

**Formal analysis:** Farina Sultan, Reelina Basu, Divya Murthy, Kuldeep S. Attri, Ayush Aggarwal, Pooja Kumari, Pooja Dnyane, Chetan Gadgil.

**Funding acquisition:** Vivek T. Natarajan, Rajesh S. Gokhale.

**Investigation:** Farina Sultan, Reelina Basu, Divya Murthy, Manisha Kochar.

**Methodology:** Farina Sultan, Kuldeep S. Attri, Pooja Kumari, Archana Singh, Neel Sarovar Bhavesh, Pankaj K. Singh.

**Project administration:** Rajesh S. Gokhale.

**Resources:** Vivek T. Natarajan, Rajesh S. Gokhale.

**Supervision:** Rajender K. Motiani, Vivek T. Natarajan, Rajesh S. Gokhale.

**Validation:** Farina Sultan, Manisha Kochar, Jyoti Tanwar.

**Visualization:** Ayush Aggarwal.

**Writing – original draft:** Farina Sultan.

**Writing – review & editing:** Farina Sultan, Vivek T. Natarajan, Rajesh S. Gokhale.

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
