## [Editor Report · Decision Letter 0]

7 Dec 2021

Dear Dr Gokhale, 

Thank you for submitting your manuscript entitled "Temporal resolution of melanogenesis determine fatty acid metabolism as key skin pigment regulator" for consideration as a Research Article by PLOS Biology, and thanks again for your patience during our initial assessment of your manuscript.

Your manuscript has now been evaluated by the PLOS Biology editorial staff, as well as by an academic editor with relevant expertise, and I am writing to let you know that we would like to send your submission out for external peer review.

Once your full submission is complete, your paper will undergo a series of checks in preparation for peer review. Once your manuscript has passed the checks it will be sent out for review. To provide the metadata for your submission, please Login to Editorial Manager (https://www.editorialmanager.com/pbiology) within two working days, i.e. by Dec 09 2021 11:59PM.

If your manuscript has been previously reviewed at another journal, PLOS Biology is willing to work with those reviews in order to avoid re-starting the process. Submission of the previous reviews is entirely optional and our ability to use them effectively will depend on the willingness of the previous journal to confirm the content of the reports and share the reviewer identities. Please note that we reserve the right to invite additional reviewers if we consider that additional/independent reviewers are needed, although we aim to avoid this as far as possible. In our experience, working with previous reviews does save time. 

If you would like to send previous reviewer reports to us, please email me at lsmith@plos.org to let me know, including the name of the previous journal and the manuscript ID the study was given, as well as attaching a point-by-point response to reviewers that details how you have or plan to address the reviewers' concerns. 

Given the disruptions resulting from the ongoing COVID-19 pandemic, please expect some delays in the editorial process. We apologise in advance for any inconvenience caused and will do our best to minimize impact as far as possible.

Kind regards,

Lucas

Lucas Smith

Associate Editor

PLOS Biology

lsmith@plos.org

---

## [Decision Letter · Decision Letter 1]

3 Feb 2022

Dear Dr Gokhale,

I am writing on behalf of my colleague Dr Lucas Smith, who is currently on paternity leave. 

Thank you for submitting your manuscript "Temporal resolution of melanogenesis determine fatty acid metabolism as key skin pigment regulator" for consideration as a Research Article at PLOS Biology. Your manuscript has been evaluated by the PLOS Biology editors, by an Academic Editor with relevant expertise, and by two independent reviewers.

In light of the reviews (below), we will not be able to accept the current version of the manuscript, but we would welcome re-submission of a much-revised version that takes into account the reviewers' comments. A third reviewer had also agreed to review your study but is now very delayed. In order not to postpone any longer communicating the decision to you, we decided to move ahead with the feedback we have in hand. If the third reviewer submits their comments later, I'll forward those to you. Please accept my apologies for the delay in sending this letter to you.

Please note that we cannot make any decision about publication until we have seen the revised manuscript and your response to the reviewers' comments. Your revised manuscript is also likely to be sent for further evaluation by the reviewers.

We expect to receive your revised manuscript within 3 months. 

**IMPORTANT - SUBMITTING YOUR REVISION**

Your revisions should address the specific points made by each reviewer with additional data and analyses where requested. Please submit the following files along with your revised manuscript:

*Re-submission Checklist*

*Published Peer Review*

*PLOS Data Policy*

*Blot and Gel Data Policy*

Sincerely,

Gabriel Gasque on behalf of

Lucas Smith

Associate Editor

PLOS Biology

lsmith@plos.org

REVIEWS:

Reviewer #1, Yejing Ge: Conducting transcriptional and metabolomic profiling on a cellular pigmentation model, Sultan et al described three dynamic phases of melanogenesis, including an initial high-MITF state with rapid uptake of glucose, followed by a pigmented state accompanied by anabolic pathways and melanosome biogenesis, and a final recovery phase presumably with activation of the NRF2 detoxication pathway. During pigmented state, the authors suggest SREBF1-mediated upregulation of fatty acid synthesis results in a transient accumulation of lipid droplets and enhancement of fatty acids oxidation through mitochondrial respiration. While this heightened bioenergetic activity is important to sustain melanogenesis, it impairs mitochondria and shifts the metabolism towards glycolysis. Linking this finding to hyper-pigmentary diseases, it was further shown that inhibitors of lipid metabolism can resolve hyper-pigmentary conditions in a guinea pig UV-tanning model. Overall, the findings are novel and interesting, the evidence presented is sufficient to support conclusion, and the manuscript is well-written. A few comments below:

Besides SREBPs, PPARs are known master regulators of lipid metabolism, particularly fatty acids synthesis and oxidation. The authors are encouraged to examine and/or discuss PPARa/b/g involvement in melanogenesis.

RNAseq and network computational analysis should be complemented by experimental validations, including WB and/or qPCR of differentially expressed genes and upstream transcriptional regulators.

Even though the Gokhale group has well-established the low-density B16 pigmentation model in previous publications (Natarajan VT et al Proc Natl Acad Sci. 2014; Motiani RK et al EMBO J. 2018), it would be beneficial to provide more details about this assay. 

Throughout the manuscript, quantification of WB is recommended to reflect dynamic changes of melanogenesis proteins.

Reviewer #2: In this paper, the authors use gene expression profiling and metabolic tracing to establish a key role for fatty acids in melanogenesis. This is a useful contribution to the field as the metabolic inputs into melanocyte differentiation are not well understood. They use a variety of pharmacologic agents to inhibit fatty acid biogenesis or metabolism to confirm their gene expression results, with limited genetic confirmation. While some of their results are correlative, it opens up downstream genetic experiments to test the hypotheses raised here. 

Major points:

1. Using both isotope tracing (Fig 3E) and Seahorse (Fig 4A-C) the authors suggest that D5 cells will increase beta-oxidation of exogenously provided fatty acids to fuel the TCA cycle. However, for much of the paper they focus on the role of fatty acid synthesis rather than uptake in providing the substrates for beta-oxidation. Could the authors explore 

a. The expression of known FA transporters in their RNA-seq data and how that compares to the expression of FA synthesis enzymes

b. Is there evidence of altered fatty acid uptake (i.e. BODIPY uptake in D5 cells) versus increased synthesis rates (i.e. labelled tracing experiments)? A FAO Seahorse assay with Palmitate could also be used to determine the contribution of extrinsic v. endogenous FAs to beta-oxidation in D5 melanocytes. 

2. In Figure 7A-E the authors show that inhibition of either FA synthesis or LD biogenesis will lead to decreased pigmentation. However, treatment with the inhibitor orlistat is thought to also inhibit FASN (similar to C75) but also block lipolysis leading to a failure to reduce LD content in Figure S5. However, melanin levels still decrease with orlistat treatment. Could the authors address whether lipolysis of lipid droplets can also affect pigmentation (for instance using Atglistatin) or if blocking FA synthesis is all that is required to decrease melanin content. 

3. In Figure 7G the representative images chosen for D0 make it appear as though even prior to treatment start pigmentation is increased in the lower half of the animal. Since the most critical comparison would be between UV treatment with and without Orlistat, and it appears that the Orlistat + UV treated group is less pigmented at baseline could the authors provide data across treatment groups such that it is clear there is a change in pigmentation regardless of the physical location of the treatment group on the animal. Alternatively the authors could remove this in vivo data since not critical to the paper. 

4. Figure 4 shows that at D6 the cells have a significant increase in damaged mitochondria. Could the authors provide additional data/explanation of why this might be the case, is this typical of highly differentiated cells? Could mitochondrial fragmentation be due to a decrease in cellular fitness, i.e. increased cell death/senescence? 

5. Figure 6 relies heavily on the use of 25-HC to block srebf1, however, 25-HC is known to have other effects outside of srebp inhibition. While specific pharmacologic inhibition of srebf1 may be difficult could the authors comment on the specificity of 25-HC for srebf1 in the text. Futhermore, are there other ways to more specifically target this pathway, for instance PPAR inhibitors?

Minor points: 

1. Could the authors provide additional explanation of the pigmentation model that they chose. For instance, a schematic showing the assay setup and temporal resolution of pigmentation changes.

2. In Figure 1D the authors do not define how they performed differential gene expression analysis. Could they state which comparisons they used to generate the fold changes represented in the heat map. 

3. In Figure 1G-H statistical analysis of the significance of LD size/number is needed. This would also support whether the claim on lines 165-166 regarding rapid depletion of LDs at D6 is true. 

4. Could the authors provide clearer data on why they chose to focus on the presence of lipid droplets as a facet of fatty acid metabolism; specifically which genes known to be involved in LD biogenesis are upregulated in D5/D6 cells? 

5. Inclusion of Figure 1I-J is confusing in this location. The data might be better suited to be presented when the authors examine the effect of the DGAT inhibitor as well as FA synthesis inhibitors on melanin content in Figure 7.

---

## [Decision Letter · Decision Letter 2]

24 Mar 2022

Dear Dr Gokhale,

Thank you for submitting your revised Research Article entitled "Temporal resolution of melanogenesis determine fatty acid metabolism as key skin pigment regulator" for publication in PLOS Biology. I have now obtained advice from the two original reviewers and have discussed their comments with the Academic Editor. 

Based on the reviews (attached below), we will probably accept this manuscript for publication, provided you satisfactorily address the remaining points raised by Reviewer 2. Please also make sure to address the following data and other policy-related requests and revise the English in the Abstract and the rest of the manuscript to correct any gramatical mistakes.

In addition, we would like you to consider a suggestion to improve the title:

"Temporal analysis of melanogenesis identifies fatty acid metabolism as key skin pigment regulator"

We expect to receive your revised manuscript within two weeks. 

*Published Peer Review History*

*Press*

Sincerely,

Ines

--

Ines Alvarez-Garcia, PhD,

Senior Editor

PLOS Biology

ETHICS STATEMENT:

-- Thank you for including the ethics statement. Please add the protocol/permit/project license you have use. Please also include an approval number.

Fig. 1A, C, D, G, H; Fig. 3C, E; Fig. 4A-C, E, F; Fig. 5B, E, F; Fig. 6C, D, H, J; Fig. 7C, E, H-J; Fig. S1A, B, F, H, J; Fig. S2A-F; Fig. S3A-F; Fig. S4B; Fig. S5B, C; Fig. S6A-C; Fig. S7A-C and Fig. S8A, B

**In addition, you should make the data deposited at the Genome Sequence Archive publicly available before the manuscript enters production (GSE164375).

Reviewers' comments

Rev. 1:

The revised manuscript has fully addressed my concerns and is acceptable for publication.

Rev. 2:

Overall, the authors have addressed most of our initial comments. A few small remaining points:

1) The authors reference the HSL data but do not include the data in the paper although it is in the methods. This data should be included and discussed.

2) The pigmentation effect of srebf1 knockdown is very modest and not really quantified. It seems like the effect of this gene alone is much smaller than the inhibitor. This should be discussed in the discussion - is it compensation? Ineffective knockdown?

---

## [Editor Report · Decision Letter 3]

19 Apr 2022

Dear Dr Gokhale,

On behalf of my colleagues and the Academic Editor, Heather Christofk, I am pleased to say that we can in principle accept your Research Article entitled "Temporal analysis of melanogenesis identifies fatty acid metabolism as key skin pigment regulator" for publication in PLOS Biology, provided you address any remaining formatting and reporting issues. These will be detailed in an email that will follow this letter and that you will usually receive within 2-3 business days, during which time no action is required from you. Please note that we will not be able to formally accept your manuscript and schedule it for publication until you have completed any requested changes.

PRESS

Sincerely, 

Ines

--

Ines Alvarez-Garcia, PhD 

Senior Editor 

PLOS Biology
